

# Development of a formalism for computing in situ transits of Earth-directed CMEs. Towards a forecasting tool II

Pedro Corona-Romero[1,2] and Pete Riley[3]

[1]Space Weather National Laboratory (LANCE), Insituto de Geofisica Unidad Michoacan, Universidad Nacional Autonoma de Mexico, Campus Morelia, Morelia, Michoacan, Mexico.
[2]CONACYT-Insituto de Geofisica Unidad Michoacan, Universidad Nacional Autonoma de Mexico, Campus Morelia, Morelia, Michoacan, Mexico.
[3]Predictive Science Inc. 9990 Mesa Rim Rd Suite 170, San Diego CA 92127, USA.

**Correspondence:** P. Corona-Romero (p.coronaromero@igeofisica.unam.mx)

**Abstract.** Earth-directed coronal mass ejections (CMEs) are of an important interest for space weather purposes, because they are precursors of the major geomagnetic storms. The geoeffectiveness of a CME mostly relies on its physical properties like magnetic field and speed. There are multiple efforts in the literature to estimate *in situ* transit profiles of CMEs, most of them based on numerical codes. In this work we present a semi-empirical formalism to compute *in situ* transit profiles of Earth-directed fast halo CMEs. Our formalism combines analytic models and empirical relations to approximate CME properties as would be seen by a spacecraft near the Earth's orbit. We use our formalism to calculate synthetic transit profiles for 10 events, including the *Bastille day event* and three *varSITI Campaign* events. Our results showed qualitative agreement with *in situ* measurements. Synthetic profiles of speed, magnetic intensity, density and temperature of protons had average errors of 10%, 27%, 46% and 83%, respectively. Additionally, we also computed the travel time of CME centers, with an average error of 9%. We found that compression of CMEs by the surrounding solar wind significantly increased our uncertainties. We also outline a possible path to apply this formalism into a space weather forecasting tool.

## 1 Introduction

According the National Space Weather Program Strategic Plan, "space weather refers to conditions on the Sun and in the solar wind, magnetosphere, ionosphere, and thermosphere that can influence the performance and reliability of space-borne and ground-based technological systems and can endanger human life or health" (Moldwin, 2008). Space weather at Earth may potentially decrease, or even stop, the operation of infrastructure, facilities, technology and services in which our society relies on (see Weaver and Murtagh, 2004). Its negative effects may compromise the distribution of energy, damage satellites components and degrade their orbits, cause malfunctions in navigation and positioning systems, as well as disrupt radio communications on Earth and in space (Echer et al., 2005a; Goodman, 2005; Kamide and Chian, 2007; Moldwin, 2008; Schrijver,





2015). Space weather perturbations are commonly due to phenomena derived from solar activity like coronal mass ejections (CMEs). Interplanetary counterparts of CMEs are closely related with major perturbations of Earth's space weather like geomagnetic storms, ionospheric disturbances, and geomagnetic induced currents (Howard, 2014). Here, we use CME to refer to coronal mass ejections whether they are remotely observed between the Sun and Earth or are directly detected in situ.

CMEs are energetic phenomena that involve the release of material, energy, and magnetic field from the solar corona into the interplanetary (IP) medium. CMEs are commonly related with other solar phenomena like solar flares, interplanetary shock waves, and others (Echer et al., 2005b; Forsyth et al., 2006). It is well known that supermagnetosonic (fast) CMEs are one of the most important triggering of intense geomagnetic storms (Ontiveros and Gonzalez-Esparza, 2010, and references therein). This condition makes CMEs a hazard for the stability of Earth's space climate and turns the capability to forecast fast-CME

arrivals into a topic of significant importance for shielding our society (Schrijver, 2015).

  The physical characteristics of CMEs are crucial for space weather purposes because they may influence the geoeffectiveness of CMEs; with the speed and inner magnetic field the most relevant (see Gonzalez et al., 2001; Xie et al., 2006; Echer et al., 2008). There have been a number of attempts to understand and describe the physical characteristics of CMEs in the inner heliosphere and beyond. Bothmer and Schwenn (1998), Liu et al. (2005), Wang et al. (2005), and Leitner et al. (2007),

empirically found tendencies to describe the physical properties of CMEs like density, magnetic field, radius, and temperature as functions of the heliocentric distance. Moreover, Gulisano et al. (2010, 2012) used an analytic approach, complemented by *in situ* data, to describe the evolution of magnetic field, radius and expansion rates of CMEs.

  Improvements in numerical codes increase their ability to mimic *in situ* data. At present, it is possible to systematically forecast the conditions of solar wind at Earth's orbit through combination of numerical, empirical and analytic models. An

example is the automated WSA + ENLIL model (Pizzo et al., 2011) used by the Space Weather Prediction Center of NOAA (http://www.swpc.noaa.gov/products/wsa-enlil-solar-wind-prediction), which combines the "ENLIL" MHD numerical code (Odstrcil, 2003), and the WSA semi-analytic model (Wang and Sheeley, 1990). The WSA model approximates the boundary values of the solar wind which are used by ENLIL to simulate the solar wind evolution out to Earth's orbit. This model can also simulates propagation of CMEs through the "ice cream cone" empirical model (Xie et al., 2004). Although numerical codes

are robust tools for space weather studies and forecasting, many issues remain (see discussion in Riley et al., 2012; Vourlidas et al., 2019).

  Analytic approaches can be useful for calculating synthetic *in situ* transit profiles of CMEs. Démoulin et al. (2008), depart from self-similar expansion hypothesis, to obtain a theoretical frame to describe *in situ* observed CME velocities. This analytic description allowed them to approximate the speed profiles during *in situ* transit profiles of CMEs. Savani et al. (2015) com-

bined statistical results of CME helicity near the Sun and a simplified flux rope solution to forecast the *in situ* magnetic field inside CMEs. This was done by extrapolating ("projecting") the initial statistically expected magnetic polarity and trajectory of the flux rope. This straightforward semi-empirical method may, in the future, be useful as a space weather forecasting tool, as Savani et al. (2017) remarked.

  Our present work complements and builds on these previous studies by estimating synthetic transit profiles of Earth-directed

fast CMEs. This work is the third study in a series aimed at developing an early alert system for arrivals of fast CMEs/plasma





sheaths/shocks. In the first paper (Corona-Romero and Gonzalez-Esparza, 2016) we presented a semi-empirical formalism to calculate *in situ* synthetic transit profiles of plasma sheaths and forward shocks, both associated with the arrival of fast CMEs. Such a formalism combined the piston-shock model (Corona-Romero and Gonzalez-Esparza, 2011, 2012; Corona-Romero et al., 2013) and the jump relations for plasmas (Petrinec and Russell, 1997) to calculate the speed, density, magnetic field,

and temperature of plasma sheaths during a CME/shock *in situ* transit. Meanwhile, in the second paper (Corona-Romero et al., 2017), we probed the capability of the piston-shock model to forecast arrival speeds and travel times of CMEs, as well as to estimate CME masses.

In order to complement our previous works, we now present a formalism for calculating synthetic transit profiles of fast CMEs. During this work we will assume that: *i*) The trajectory of the CME leading edge and its mass are well approximated

by the piston-shock model; *ii*) CMEs have a croissant-like geometry of constant angular width with a radius that follows a self-similar expansion; *iii*) The cylinder radius is significantly shorter than the distance between the Sun and the cylinder center; *iv*) The CME mass is constant, homogeneously distributed and can be described as a polytropic plasma; and *v* The CME magnetic field is a force-free flux rope.

In the next sections we combine the piston-shock model and empirical relations to analytically describe the trajectories and

total mass of CMEs as a whole (Section 2.1). Subsequently, in Section 2.2, we present the relations for calculating the synthetic transit profiles of CMEs. In Section 3 we test our formalism by calculating synthetic transit profiles for 10 Earth-directed fast CMEs. After, in Section 4, we discuss our results, as well as the power and limitations of our formalism. Finally, we present our general conclusions.

## 2   Formalism to compute synthetic transits of CMEs

In order to present our formalism to compute synthetic transits of CMEs, in Section 2.1 we describe the way we implement the piston-shock model to approximate the trajectory (position and speed) of the CME as whole. In Section 2.2, we analyse an event to introduce the expressions to estimate the synthetic transit profiles of CMEs.

### 2.1   An analytic model for CME propagation

The CME trajectories calculated by the piston-shock model have two phases: a short interval of constant speed followed by a

period where the CME speed asymptotically approaches the speed of the solar wind. Previous studies suggested that the first phase ends around $30\,R_\odot$, hence the deceleration phase dominates CME propagation up to the orbit of the Earth ($d_\oplus = 1\,AU$) (Corona-Romero et al., 2013, 2015). During the deceleration phase, the position ($L$) and speed ($\dot{L}$) of the leading edge of the CME is given by:

$$\frac{L(t)}{u_1\,\tau_f} = \frac{L_0}{u_1\,\tau_f} + \left(\frac{t}{\tau_f} - a\,c\right) + \left[2\,a\,c\,(a-1)\,\frac{t}{\tau_f} - a^2\,c\,(1-c)\right]^{1/2} \tag{1}$$



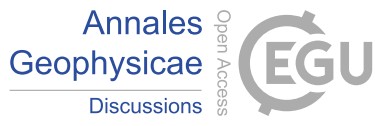

and

$$\frac{\dot{L}(t)}{u_1} = 1 + (a-1)\left[\frac{ac}{2(a-1)\frac{t}{\tau_f} - a(1-c)}\right]^{1/2},$$  (2)

respectively, where $t$ is time ($t \geq 0$), and $L_0$ and $\dot{L}_0$ are the initial ($t = 0$) position and speed of CME leading edge. It is important to remark that $\dot{L}_0$ is the speed value during the constant-speed phase. Additionally, $u_1$ is the *in situ* solar wind bulk speed and $\tau_f$ the rising phase duration (Zhang and Dere, 2006) of the associated solar flare. The constants $a$ and $c$ are

non-dimensional, and are related to the inertia of the CME. The constant $a$ is given by:

$$a = \frac{\dot{L}_0}{u_1} + \frac{1}{\sqrt{c}}\left(\frac{\dot{L}_0}{u_1} - 1\right).$$  (3)

While $c$ is treated as a free parameter to match the calculated arrival time with its *in situ* registered counterpart.

To accurately reconstruct the trajectory of a CME as a whole, we need to specify the shape of the CME. As an initial approximation, we can assume that CME shapes are croissant-like (see Figure 1). Thus, we can approximate the CME core as

a cylinder that contains most of the CME material (shaded region in Figure 1). It is important to note that while the geometry we use in this work is more suitable for magnetic clouds or, more recently, the so called "flux-rope CMEs" (see Vourlidas et al., 2013), these procedures can be adapted to any simple geometry.

It is also important to remark that the transverse section of the CME gradually deform from almost-circular, in solar corona, into a "pancake" shape in the IP medium. Such a geometrical change is due to a non-homogeneous expansion (see Riley et al.,

2004). Therefore, assuming a circular transverse section is a roughly approximation which could be preferably suitable for the central portions inside flux ropes.

It is believe that the radius of CMEs ($R$) follows a self-similar expansion in the IP medium (e.g. Liu et al., 2005; Wang et al., 2005; Leitner et al., 2007). In fact, there is evidence of self-similar growth of CME radius even in the solar corona (Mierla et al., 2011). Hence, in this work, we also assume that $R$ obeys the empirical relation found by Bothmer and Schwenn (1998)

and later verified by Gulisano et al. (2010):

$$\frac{R}{d_\oplus} = 0.12\,k\left(\frac{r}{d_\oplus}\right)^\epsilon,$$  (4)

where $\epsilon = 0.78 \pm 0.12$, and $r$ the CME center (see Figure 1). We introduce in Equation (4) the non-dimensional constant $k$, which is a free parameter used to express wider ($k > 1$) or thinner ($0 < k < 1$) CMEs than the defined average ($0.12\,d_\oplus$). It is important to remark that the value of $\epsilon$ is not fixed and it can change according the solar wind conditions in which a CME

expands (see Gulisano et al., 2010). In this work we use a representative value, and the way we present our equations allows us to easily use another value.

More generally, we can express the CME center ($r$) as:

$$r = L - R.$$  (5)

Since $R < L$, we can combine Equations (4) and (5) and expand the result up to second order around $L$. The result is:

$$R = \frac{1 + R'_L - \sqrt{(1 + R'_L)^2 - 2R_L R''_L}}{R''_L},$$  (6)





where we have used $R_L = R(L)$. Additionally, $R'_L = \epsilon R_L/L$ and $R''_L = \epsilon(\epsilon - 1)R_L/L^2$ are the first and second derivatives of $R_L$, respectively. By combining Equations (6) and (5) we can express the CME center position as function of $L$, *i.e.* Equation (1).

Taking the time derivative of Equation (6), we obtain the expansion speed of CMEs ($\dot{R}$):

$$\frac{\dot{R}}{\dot{L}} = 1 - \frac{R'''_L}{(R''_L)^2}\left(R'_L + 1\right)$$
$$+ \frac{(1 + R'_L)^2 R'''_L - (R''_L)^2 - R_L R''_L R'''_L}{(R''_L)^2 \sqrt{(1 + R'_L)^2 - 2R_L R''_L}}. \tag{7}$$

With $R'''_L = \epsilon(\epsilon - 1)(\epsilon - 2)R_L/L^3$, being the third derivative of $R_L$. It follows that the speed of the CME center ($\dot{r}$) is given by the time derivative of Equation (5):

$$\dot{r} = \dot{L} - \dot{R}. \tag{8}$$

Again, by combining Equations (2), (7), and (8) we can express the CME center speed through the speed of the CME leading edge. Additionally, we can estimate the travel time of the CME axis ($TT_r$), or CME "center", by:

$$\frac{TT_r}{\tau_f} = a^2 c + \frac{d_\oplus + R_\oplus - L_0}{u_1 \tau_f}$$
$$- \sqrt{2ac(a-1)\left[\frac{d_\oplus + R_\oplus - L_0}{u_1 \tau_f}\right] + a^2 c(a^2 c - 1)}; \tag{9}$$

where $R_\oplus = R(d_\oplus)$. Equation (9) was obtained by solving Equation (1) for the condition $L = d_\oplus + R$.

Once the CME center position and radius are known, the piston-shock model allows us to calculate the CME mass, which depends on the initial conditions and shape of the CME (see discussion in Corona-Romero et al., 2017). For simplicity, if we assume the CME mass uniformly distributed within its volume, we can express the CME density ($\rho$) as:

$$\rho = ac\, m_p\, n_1 u_1 \tau_f \left[\frac{\theta_0 d_\oplus^2 R_0}{2\theta r_0 r R^2}\right], \tag{10}$$

where $n_1$ is the *in situ* solar wind proton density, $m_p$ is the proton mass, and $\theta$ is the semi-angular width of CMEs; additionally the index "0" denotes initial values (at $t = 0$). It is important to note that in Equation (10) we also assume the CME mass is conserved, a condition that might be violated when significant magnetic reconnection occurs between the CME and its surrounding solar wind (*e.g.* Dasso et al., 2007).

## 2.2 Calculating *in situ* transit profiles of CMEs

Next, we present our procedure for calculating the synthetic transit profiles of CMEs. For simplicity, we use an astronomical unit to compute our synthetic transits; however our equations could be easily adapted for another heliocentric distances. We also assume that the spacecraft (*i.e.*, Earth) and the trajectory of the CME center are almost aligned, that is, the spacecraft crosses near the CME center. This simplification allows us to neglect projection effects as a first approximation, but limits our





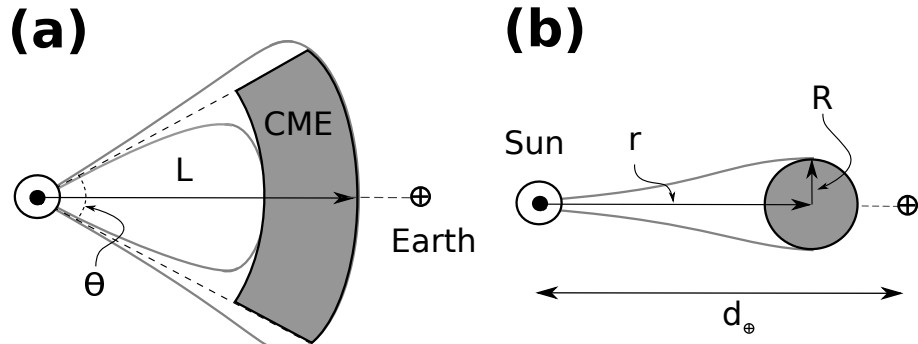

**Figure 1.** Sketch for the croissant-like geometry (thick solid grey line) of CMEs assumed in this work. Panels *(a)* and *(b)* show a meridian and equatorial view of the CME, respectively. We approximate the CME material through a cylinder (shaded region). In the panels we present the locations of leading edge ($L$), center ($r$) and radius ($R$) of CME and its semi-angular width ($\theta$). We also present the position of the Sun ($\odot$) and the Earth ($\oplus$) as references.

formalism to CMEs whose source region is located near the center of the solar disc. We leave the solution of a more general scenario for future studies.

We will use event 1 from Table 1 to illustrate the steps of our formalism. Figure 2 shows the *in situ* measurments (solid and dotted black lines) during the transit of event 1 past Earth. From top to bottom the panels show the magnitude of solar wind radial speed ($|V^x|$), density ($N_p$) and temperature ($T_p$) of protons, and magnetic field magnitude ($B$). On the left-most portion of all panels we observe ambient solar wind up to the shock arrival (20000608-09:10), which is a spontaneous jump in all *in situ* measurements. After the shock comes the solar wind perturbed by the shock (sheath) and, behind it, the CME. We note

that during the CME transit the plasma-$\beta$ (gray solid line in $N_p$ panel) significantly decreases, and the value of $T_p$ is lower than the expected temperature of protons (gray solid line in $T_p$ panel). Following the CME, there is again ambient solar wind.

Our first step is to measure the travel time ($TT$) spent by the CME leading edge in traveling from near the Sun (reported detection time) to Earth's orbit (*in situ* detection). We mark $TT$ on the left side of the panels in Figure 2 by a vertical dotted red line. With the value of $TT$ known, we proceed to find the value of $c$ (through Equation (1)) that makes $L(TT) = d_\oplus$.

The values of $L_0$ and $\dot{L}_0$ we used were the reported initial position and speed by CME LASCO Catalog (Yashiro et al., 2004; Gopalswamy et al., 2009), respectively. Additionally, the horizontal solid green lines in the panels of Figure 2 mark out the solar wind values used in our calculations; values taken around 18-10 hours before the CME's arrival. Table 1 list the input values used in the analysis of Event 1.

The second step is to measure the time required for the CME to cross the Earth's orbit, *i.e.*, the transit time ($\Delta T$). In the

upper panel of Figure 2 we enclosed $\Delta T$ with dotted red lines; the left line marks the CME arrival, whereas right line marks the trailing edge of the CME. Hence, at the time $t = TT + \Delta T$ the separation between the CME leading edge and Earth's orbit would be $2R$. Thus, after combining and manipulating Equations (1) and (4), we obtain:

$$k = \left( \frac{L(TT + \Delta T) - d_\oplus}{0.24\, d_\oplus} \right) \left( \frac{2\, d_\oplus}{L(TT + \Delta T) + d_\oplus} \right)^\epsilon . \tag{11}$$



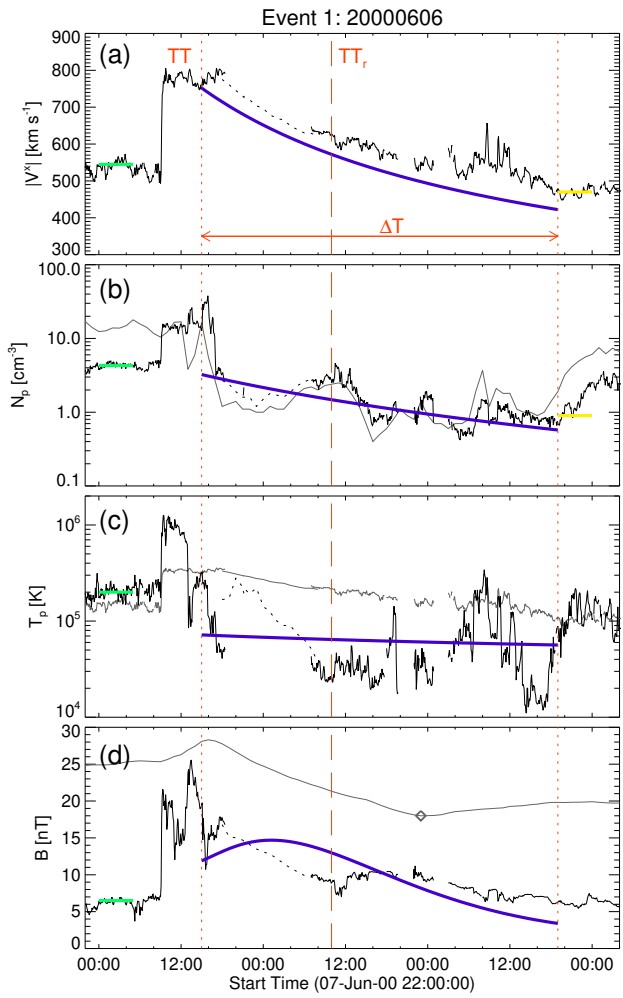

**Figure 2.** Calculated synthetic transit of event 1. From top to bottom, the panels *(a)*, *(b)*, *(c)*, and *(d)* present the radial component of the solar wind speed ($|V^x|$), the density ($N_p$) and temperature ($T_p$) of protons and the magnetic field intensity ($B$), respectively. Solid blue lines show our model results and the solid black lines are *in situ* measurements as extracted from NASA/GSFC's OMNI data set through OMNIWeb service. Dotted and dashed vertical red lines mark the CME boundaries and center, respectively. Green solid lines mark *in situ* solar wind used for calculations and solid yellow lines mark solar wind behind the CME (see Table 1). Solid grey lines in $N_p$ and $T_p$ panels are the plasma Beta (10 folded) and the expected proton temperature ($T_{exp}$) (Lopez, 1987), respectively. Solid grey line in panel (d) is the accumulative magnetic flux, as defined by Dasso et al. (2006), whose extremum (open diamond) gives an estimation for the magnetic center inside the CME.

Since we already know the values of $TT$ and $\Delta T$, Equation (11) allows us to compute the value of the free parameter $k$, for a
given value of $\epsilon$.





Once the values of the free parameters $c$ and $k$ are known, in our third step we compute the trajectory (Equations (1) and (2)), radius (Equation (6)) and expansion rate (Equation (7) of the CME during the period $TT < t < \Delta T + TT$. Following this, we can express the speed ($V^x$) on the Sun-Earth line that would be "observed" at *in situ*:

$$
\begin{aligned}
V^x &= \dot{r} + \left(\frac{d_\oplus - r}{R}\right)\dot{R} \\
&= \dot{L} + \left(\frac{d_\oplus - L}{R}\right)\dot{R}. 
\end{aligned} \tag{12}
$$

In Equation (12) we assumed that the velocity of CME material linearly grows with the radial distance from the CME axis (see Démoulin and Dasso, 2009). We overplot our calculated *in situ* speed (solid blue line) on the upper panel of Figure 2. We note that our calculated speed closely follows its measured counterpart; however, the synthetic profile is below the *in situ* data. This issue might be fixed by using values of $\dot{L}_0$ calculated by multi-spacecraft instead of using coronagraph images from one spacecraft. This is because speeds are underestimated by simple coronagraph images due to projection effects.

The fourth step consists of calculating the density and temperature profiles. Since the CME mass is homogeneously distributed, the density of protons ($N_p$) seen at *in situ* location is expressed by:

$$
N_p = \left[\frac{acn_1u_1\tau_f}{1 + 4q_\alpha}\right]\left[\frac{\theta_0 d_\oplus^2 R_0}{2\theta r_0 r R^2}\right]. \tag{13}
$$

In the last expression, we depart from Equation (10) by assuming an average ratio $q_\alpha$ between alpha particles and protons inside the CME. Additionally, for simplicity, we assumed a constant value for $\theta$ and a content of $12\%$ fraction of alpha particles in the CME material (Borrini et al., 1982; **?**). Since we assume the CME material to be a polytropic gas, we can express the temperature of protons ($T_p$) by combining Equation (10) and the well known expression for the temperature of a polytropic gas with polytropic index $\gamma$, and, after some manipulation:

$$
\frac{T_p}{T^*} = 35401\,\mathrm{K}\left[\frac{N_p}{N_{p1}}\right]^{(\gamma-1)}, \tag{14}
$$

where $N_{p1}$ is the CME proton density at $r = d_\oplus$ and $T^*$ a free parameter that indicates if the CME is hotter ($T^* > 1$) or colder ($T^* < 1$) than the approximated average temperature ($35401\,\mathrm{K}$) in CMEs (see Liu et al., 2005). We selected the value of $T^*$ that allowed the median of Equation (14) to match the median of *in situ* temperature during $\Delta T$.

Regarding the polytropic behaviour of CME material, a theoretical approach by Chen and Garren (1993) showed that an adiabatic expansion ($\gamma = 5/3$) of flux ropes may derive into temperatures lower than expected. This work was followed by others that used $1 < \gamma < 5/3$ for studying magnetic clouds (e.j. Gibson and Low, 1998; Chen, 1996; Krall et al., 2000). After, Liu et al. (2005) studied statistical properties of CMEs, one of those properties was the thermodynamics of CMEs, finding that the $\gamma = 1.14 \pm 0.03$, value that we use in our calculations. Once more, we present our equations in such a way that facilitates the usage of a value of $\gamma$ different to the one we use.

It is widely know that the Lundquist (1951) solution of a stationary flux rope's magnetic field is a useful tool to approximate magnetic fields of magnetic clouds (e.j. Burlaga, 1988, and many others). Such a solution has been extended for a number of scenarios (Vandas et al., 2006, discussed some of them). One of those extensions is the work by Shimazu and Vandas (2002)





who found that polar and axial components, and thus the magnitude, of the Lundquist solution change at the same rate for a flux rope that simultaneously expands and elongates. In addition, there is empirical evidence that indicates magnetic field intensity of CMEs decreases with the growth of the heliocentric distance (e.g. Liu et al., 2005; Leitner et al., 2007). Furthermore, such

a decrease can be approximated as a self similar relation of $r$ (e.j. Bothmer and Schwenn, 1998; Wang et al., 2005; Liu et al., 2005; Forsyth et al., 2006; Leitner et al., 2007, and others), relation that was theoretically explored by Gulisano et al. (2010).

Thus, in order to keep our expression as simple as possible, it is reasonable to locally approximate the *in situ* magnetic field magnitude of CMEs ($B$) by:

$$\frac{B}{b} = 10.9\,\text{nT} \left[\frac{r}{d_\oplus}\right]^{-1.85} \sqrt{J_0^2\left(\alpha\,\frac{|d_\oplus - r|}{R}\right) + J_1^2\left(\alpha\,\frac{|d_\oplus - r|}{R}\right)}, \tag{15}$$

where the square root in Equation (15) is the magnitude of the Lundquist solution, with $J_0$ and $J_1$ the first and second Bessel functions, respectively and $\alpha$ is the $J_0$'s first zero. In addition, the other terms on the right side of Equation (15) correspond to the empirical tendency by Gulisano et al. (2010), that controls the decaying rate of the magnetic field magnitude as heliocentric distance (time) grows larger.

Although Equation (15) may share similarities with other physics-based expressions (e.g. Farrugia et al., 1993; Cid et al.,
2002; Berdichevsky et al., 2003; Nakwacki et al., 2008; Möstl et al., 2009; Vandas et al., 2009; Mingalev et al., 2009; Nieves-Chinchilla et al., 2016, , and many others); we emphasize that such an equation is a simplified straightforward expression to estimate representative data. Nevertheless, we anticipate that Equation (15) is consistent with a particular case of the physical model by Démoulin et al. (2008), as we will discuss latter in Section 4. This point is important and, in contrast with other works, because Démoulin et al. (2008) simultaneously includes radial and axial expansions of the flux rope, as well as the
acceleration on CME bulk speed.

In Equation (15) we introduced the non-dimensional free parameter $b$ to express stronger ($b > 1$) or weaker ($0 < b < 1$) intensities of CME magnetic field, in comparison with the average value of $10.9\,\text{nT}$ (see Gulisano et al., 2010). Hence, our fifth step is to calculate the value of $b$, which value we select to minimize the average error on our calculated intensity of magnetic field:

$$\varepsilon_B = \frac{1}{N} \sum_{i=1}^{N} |B_i^{calc} - B_i^{insitu}|, \tag{16}$$

where $N$ is the number of available data points during the CME *in situ* transit, and $B^{calc}$ and $B^{insitu}$ correspond to the calculated and measured *in situ* magnetic field intensities, respectively.

Finally, our last step is to calculate the travel time associated with the CME center ($TT_r$); which is done by Equation (9) and for which the parameters are already known. The calculated moment at which the CME center transits the Earth's orbit is
shown with a vertical dashed red line in all panels of Figure 2. We compare our calculated value for $TT_r$ with the extremum (open diamond) of the the accumulative magnetic flux (solid grey line) in panel (d) (Dasso et al., 2006). It is important to note that this extremum gives an estimation for the magnetic center inside the CME.



**Table 1.** Input data for our analysis. From left to right: event number, CME detection date and time, associated active region position on the solar disk (latitude and longitude), rise time of associated solar flare, initial position and speed of CME leading edge, in-situ values of the proton density and speed of solar wind ahead (index "1") and behind (index "2") CMEs, and travel and transit times.

| | Event[a] | | Inputs[b] | | | | | | | | |
|---|---|---|---|---|---|---|---|---|---|---|---|
| | Date-Hour | Flare location | $\tau_f$ | $L_0$ | $\dot{L}_0$ | $n_1$ | $u_1$ | $n_2$ | $u_2$ | $TT$ | $\Delta T$ |
| # | [UT] | [°] | [h] | $R_\odot$ | [km s$^{-1}$] | [cm$^{-3}$] | [km s$^{-1}$] | [cm$^{-3}$] | [km s$^{-1}$] | [h] | [h] |
| 01 | 20000606-1554 | N21E10 | 0.37 | 3.98 | 1119 | 4.3 | 545 | 0.7 | 470 | 47.1 | 52.0 |
| 02 | 20000714-1054[c] | N17W11 | 0.34 | 5.21 | 1674 | 3.7 | 700 | 1.2 | 660 | 32.1 | 37.2 |
| 03 | 20010426-1230 | N16W15 | 1.33 | 4.83 | 1006 | 1.9 | 455 | 1.0 | 420 | 49.5 | 54.0 |
| 04 | 20011122-2330 | S17W24 | 0.79 | 4.77 | 1437 | 4.1 | 450 | 1.2 | 620 | 38.5 | 24.0 |
| 05 | 20031118-0850 | N03E08 | 0.34 | 6.30 | 1660 | 4.8 | 444 | 9.5 | 550 | 49.2 | 15.0 |
| 06 | 20040120-0006 | S14W10 | 1.05 | 2.90 | 965 | 4.6 | 472 | 4.5 | 530 | 58.4 | 28.5 |
| 07 | 20050513-1712 | N12E19 | 0.48 | 4.57 | 1689 | 2.8 | 415 | 0.7 | 495 | 36.8 | 52.0 |
| 08 | 20120712-1648[d] | S17E06 | 0.90 | 2.85 | 885 | 5.2 | 325 | 1.9 | 425 | 61.7 | 46.5 |
| 09 | 20140910-1800[d] | N15E14 | 0.33 | 3.75 | 1267 | 7.4 | 360 | 1.2 | 480 | 52.8 | 38.2 |
| 10 | 20150621-0236[d] | N12E16 | 0.49 | 3.53 | 1366 | 10.0 | 340 | 3.7 | 680 | 46.1 | 36.3 |

a Detection time and inputs are reported in LASCO CME Catalog (http://cdaw.gsfc.nasa.gov/CME_list/).

b Input data were acquired from LASCO CME Catalog, GOES X-ray flux (sxi.ngdc.noaa.gov/), *in situ* data by OMNIWeb (http://omniweb.gsfc.nasa.gov/). Additionally, $TT$ and $\Delta T$ were defined by *in situ* signatures of CMEs (**?**), and data from Richardson and Cane (2010) list (http://www.srl.caltech.edu/ACE/ASC/DATA/level3/icmetable2.htm).

c Case 2 is the *Bastille day event*.

d *varSITI Campaign* events (http://www.varsiti.org/).

## 3 Testing our formalism

To explore the ability of our formalism to approximate *in situ* transit profiles of CMEs, we analyzed ten Earth-directed halo
CMEs listed in Table 1. The events were selected from the LASCO Catalogue (Gopalswamy et al., 2009) and occurred during the 2000–2015 period. The objective of our selection criteria was to isolate events that fulfilled most of our formalism's assumptions and consisted of five points: 1) Fast CMEs according to coronagraph images ($\dot{L}_{cme0} > 800$ km s$^{-1}$); to ensure the effectiveness of a piston-shock approximation to model the CME trajectory; 2) CMEs associated with solar flares for which active region was located near the solar disk center, to reduce *in situ* geometrical effects on propagation and expansion speeds
of CMEs; 3) CMEs that were almost isolated (not complex) events preceded by an observed shock wave *in situ* and *in situ* signatures that were clear enough to be detected; 5) The ambient solar wind (at 1 AU) was stable enough about twelve hours before the ICME-shock arrival in order to assume an almost quiet solar wind. Table 1 lists the events studied, and the inputs used in our calculations.





**Table 2.** Results from our analysis. From the left to the right: event number, CME detection date and time, values of free parameters and associated errors to our calculated results.

| Event | | Free Parameters | | | | Associated Errors | | | | |
| --- | --- | --- | --- | --- | --- | --- | --- | --- | --- | --- |
| # | Date-Hour [UT] | $c$ | $k$ | $T^*$ | $b$ | $^a\varepsilon_{TTr}$ [h] | $^b\varepsilon_V$ [km s$^{-1}$] | $^b\varepsilon_N$ [cm$^{-3}$] | $^b\varepsilon_T$ [kK] | $\varepsilon_B$ [nT] |
| 01 | 20000606 | 12.46 | 2.80 | 2.03 | 22.9 | 13.0 | 53.3 | 0.87 | 57.1 | 2.20 |
| 02 | 20000714 | 10.95 | 2.79 | 2.22 | 46.5 | 5.4 | 47.1 | 1.40 | 55.7 | 9.14 |
| 03 | 20010426 | 4.65 | 2.69 | 0.74 | 15.9 | 0.8 | 53.1 | 1.46 | 12.3 | 2.11 |
| 04 | 20011122 | 6.52 | 1.64 | 1.45 | 26.5 | 0.9 | 57.5 | 2.03 | 54.2 | 2.25 |
| 05 | 20031118 | 2.71 | 0.88 | 3.10 | 55.9 | 2.8 | 14.6 | 10.45 | 39.9 | 7.54 |
| 06 | 20040120 | 1.75 | 1.47 | 2.48 | 18.1 | 3.3 | 34.5 | 3.18 | 46.2 | 2.44 |
| 07 | 20050513 | 8.52 | 2.93 | 1.16 | 36.0 | 4.0 | 119.5 | 1.16 | 32.0 | 8.09 |
| 08 | 20120712 | 9.30 | 1.98 | 1.32 | 29.3 | 7.0 | 55.2 | 2.75 | 30.2 | 2.70 |
| 09 | 20140910 | 11.83 | 1.84 | 1.16 | 27.3 | 12.8 | 100.8 | 2.01 | 39.8 | 5.26 |
| 10 | 20150621 | 10.70 | 1.92 | 3.37 | 19.2 | 8.9 | 70.3 | 5.24 | 138.6 | 3.55 |
| Averages | | 7.94 | 2.09 | 1.90 | 29.8 | 5.9 | 60.6 | 3.06 | 40.7 | 4.53 |

$a$ Absolute error when compared with the extreme value of magnetic flux detection-time.

$b$ Associated errors to speed ($\varepsilon_V$), density ($\varepsilon_N$) and temperature ($\varepsilon_T$) profiles are calculated with expressions similar to Equation (16), but using the values of speed, densities and temperatures instead of magnetic field.

We calculated the synthetic *in situ* profiles and CME center travel time for events 2-10 by following the procedure we
described in Section 2.2 for event 1. We present our results in Figures 4 and 5 following almost the same format used in
Figure 2. The figures show the *in situ* measurements (solid black lines) of radial speed, density and temperature of protons, and
magnetic field magnitude, as well the calculated travel time for the CME center (vertical red-dashed lines). Solid grey lines
in $N_p$, $T_p$ and $B$ panels are the plasma Beta (multiplied by 10), the expected temperature of protons (Lopez, 1987), and the
accumulative magnetic flux (Dasso et al., 2006) in arbitrary units, respsectively. At the left side of all speed and proton density
panels, we highlight the *in situ* solar wind conditions used as inputs (solid green lines), and the CME boundaries are marked
by vertical dotted red lines.

Table 2 and Figure 3 provide general insight about our results, since they present the absolute and proportional errors
associated with our calculations, respectively. It is important to highlight that in both, the table and the figure, we used the
absolute difference between our calculations and *in-situ* measurements as error, in a similar way we did for $\varepsilon_B$ (Equation (16)).
According to Figure 3, our results with lower errors are the calculated $TT_r$ (purple bars), and the synthetic transits of speed
(cyan bars) and magnetic intensity (yellow bars); with average proportional errors of 8.7%, 9.6%, and 27%, respectively.





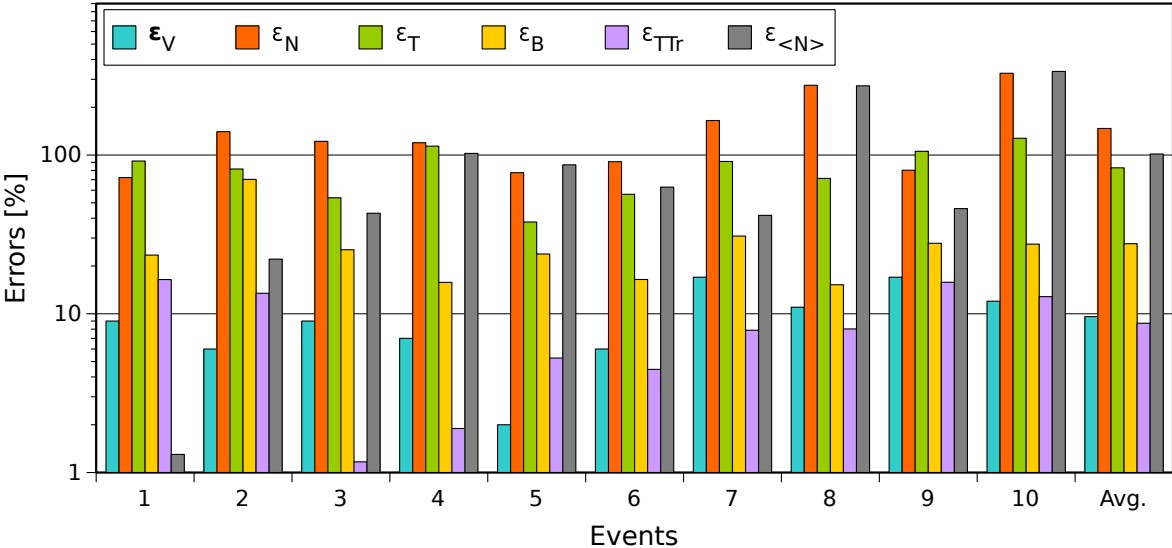

**Figure 3.** Proportional error histograms associated to our calculated synthetic profiles for speed (cyan), proton density (orange), temperature (green), and magnetic field (yellow). The last set of bars correspond to the averaged values. Additionally, grey bars are the errors when comparing the median values of proton density ($\varepsilon_{<N>}$); and purple ones correspond to the error when comparing the $TT_r$ with the transit of the accumulative magnetic field flux's extremum.

In contrast, the proportional errors for temperature (green bars) and density (orange bars) of protons were significantly larger than the first ones, with values of 83% and 143% as average, respectively. Although the errors for temperature and density are remarkably large, we found that such large errors are driven by inherent properties of the *in situ* data. For example, when we calculate the error between measured and calculated median values of proton density ($\varepsilon_{<N>}$) instead of the average error for all the data points ($\varepsilon_N$), we found that $\varepsilon_{<N>}$ drops to $\sim 101\%$. In addition, when we neglect the events affected by interacting currents of solar wind, such an error falls to 46%. We discuss our results in the next sections.

### 3.1 Synthetic profiles of speed

According to Figure 3, the calculated speed profiles accurately resemble their observed *in-situ* registered counterparts with proportional errors below 17%. Our speed results had the best performance between synthetic profiles with an average error of 61 $\mathrm{km\,s^{-1}}$ ($\sim 10\%$), which is not significant when compared with *in situ* transit-speeds of CMEs (400-1000 $\mathrm{km\,s^{-1}}$). In Figures 2, 4 and 5 we note that synthetic speed-profiles (solid blue lines) closely follow the *in situ* measurements (gray lines) for all cases. It is important to note that calculated profiles are systematically lower than their *in situ* observed counterparts. However, in the best(worst) of the cases, such a systematic underestimation derived into an average difference of 15 $\mathrm{km\,s^{-1}}$(119 $\mathrm{km\,s^{-1}}$), *i.e.*, a difference of 2.4%(17.0%), see Table 2.



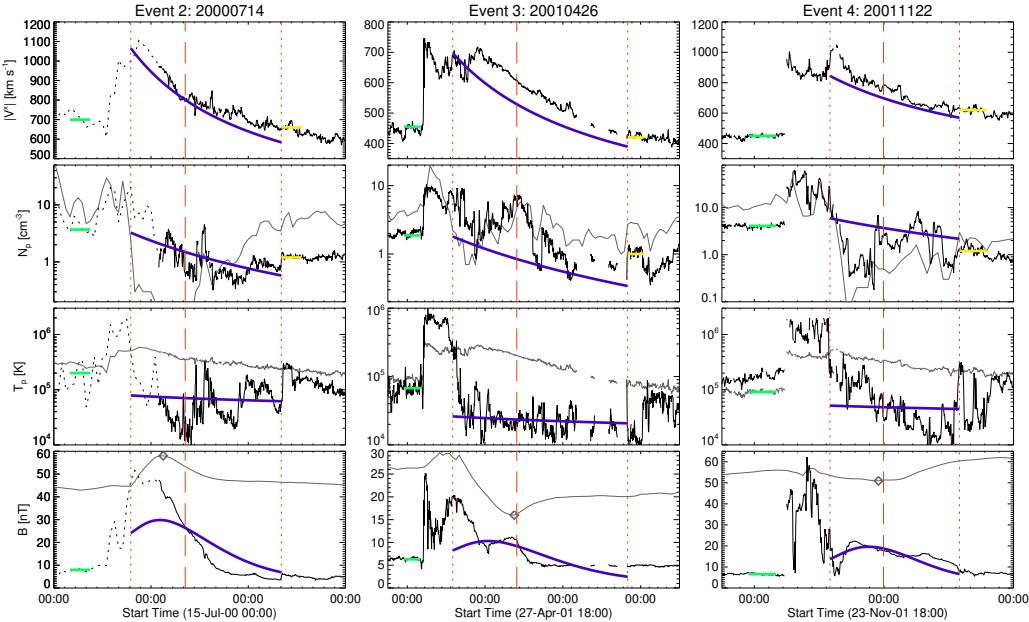

**Figure 4.** Calculated synthetic transit of events 2, 3, and 4; each column shows a different event. From top to bottom, the panels present the radial component of the solar wind speed ($|V^x|$), the density ($N_p$) and temperature ($T_p$) of protons and the magnetic field intensity ($B$). Solid blue lines are ours model results. Dotted and dashed vertical red lines mark the CME boundaries and center, respectively. Dotted red lines mark the CME boundaries. The solid black lines are *in situ* measurements as extracted from NASA/GSFC's OMNI data set through OMNIWeb service. Green solid lines mark *in situ* solar wind used for calculations (see Table 1). The solid grey lines in $N_p$ and $T_p$ panels are the plasma beta (10 folded) and the expected proton temperature ($T_{exp}$) (Lopez, 1987), respectively. The solid grey line in $B$ panel is the accumulative magnetic flux, as defined by Dasso et al. (2006), whose extremum (open diamond) gives an estimation for the magnetic center inside the CME.

It is important to note that all of our synthetic speed-profiles reproduce the speed-decreasing tendency called *aging* (Osherovich et al., 1993), commonly associated with the CME expansion. The aging effect is also present in the *in situ* data; however, it varies from one event to another; a condition that is easily observed in synthetic profiles. For example, on one hand, we have event 7 for which the speed profile decreases with a pronounced curve-like shape (see Figure 5). On the other hand,
the speed-profile of event 5 decreases almost like a line of constant slope. It is widely accepted that the difference between the 'initial' and 'final' *in situ* speed of a CME is directly related to the magnitude of its expansion speed. However, the source for the 'curve-like' or 'constant-slope' shapes is not commonly discussed. Furthermore, as is well known, to have curvature in the speed *vs* time profile requires a net acceleration; in our case, such an acceleration is related with the change (deceleration) in expansion speed ($\Delta\dot{R}$) during $\Delta T$.
The Figure 6 shows four panels related with changes in CME speeds during $\Delta T$. The upper left panel shows a histogram with the proportional changes for CME center ($\Delta\dot{r}/\dot{r}$, cyan bars) and expansion ($\Delta\dot{R}/\dot{R}$, blue bars) speeds during $\Delta T$ for all





the events, and the averages (rightmost bars). We note that, on average, the proportional changes on $\dot{r}$ (-8.2%) are small when compared with those of $\dot{R}$ (-19.2%); a condition that suggests $\Delta\dot{R}$ as a source for the curve-like shapes for speed profiles. Note also that $\Delta\dot{R}$ cover a wide range of values; where previously described events 5 and 7 are two extreme examples, with values

for $\Delta\dot{R}$ of $\sim 9\%$ and $-31.5\%$, respectively.

In the case of event 5, the value of $\Delta\dot{R}$ allow us to assume that $\dot{R}$ is almost constant during $\Delta T$, a condition that provokes the 'constant slope' shape in the speed profile of event 5 (see Figure 5) due to the absence of accelerations during $\Delta T$ ($\Delta\dot{R} \sim 0$ and $\Delta\dot{r} \sim 0$). We can verify this in Event 6 ($\Delta\dot{r} \sim 11\%$) which also shows the constant-slope speed profile (see Figure 5). In contrast to Event 5, Event 7 has a value of $\Delta\dot{R}$ (-31.5%) far above the average, with deceleration that provokes the curve-like

speed profile. We can corroborate this in other cases with high decelerations like Events 2 and 3, with values of $\Delta\dot{R} \sim -22\%$, that also present the curve-like shape (see Figure 4).

To verify the influence of $\Delta\dot{R}$ on the apparent curvature due to the aging, we examined how soon a CME center passes by the orbit of Earth. We do so by comparing the calculated transit times of the half-ahead region ($\Delta T_a$) of CMEs with their behind counterparts ($\Delta T_b$). The upper right panel of Figure 6 shows the ratio $\Delta T_a/\Delta T_b$ as a function of $\Delta\dot{R}$ (solid diamonds);

and $\Delta\dot{r}$ (open circles) for completeness. In the panel we note a tendency between $\Delta\dot{R}$ and the transit times ratio (dotted-line). In contrast, there is not a clear relation for the case of $\Delta\dot{r}$.

This tendency indicates that $\Delta T_a << \Delta T_b$ for large decelerations ($\Delta\dot{R} << 0$), and the transit times ratio gradually grows larger as $\Delta\dot{R}$ approaches to zero. The tendency suggests that, when the deceleration is negligible ($\Delta\dot{R} \sim 0$), $\Delta T_a \sim \Delta T_b$, *i.e.*, the CME center crosses the Earth's orbit almost at the midpoint of $\Delta T$, these being the conditions for constant slope speed

profile. In contrast, when $\Delta\dot{R} << 0$, the CME center crosses early, compared with $\Delta T$, at the orbit of Earth. This *early* passage of the CME center constrains all the leading-material of a CME to *rapidly* pass through the point of measurement, while forces the *delayed* trailing-material to a *slow* crossing through the Earth's orbit. These conditions became to the curve-like profiles observed for large decelerations.

Due to the importance of $\Delta\dot{R}/\dot{R}$; we examine it through the relation $\Delta\dot{R} = \ddot{R}\Delta T$. In order to do so, we depart from Equation

(4) by assuming $\Delta T \sim 2R/\dot{r}$ and evaluating for $r = d_\oplus$. After some algebra we arrive at:

$$\frac{\Delta\dot{R}}{\dot{R}} \sim -0.24(1 - \epsilon)k + \frac{\Delta\dot{r}}{\dot{r}} \sim -0.053k - 0.09. \tag{17}$$

Equation (17) explains the reason $\Delta\dot{R}$ is systematically larger than $\Delta\dot{r}/\dot{r}$ (see upper-right panel of Figure 6), since it combines two independent processes: the deceleration of bulk speed and the effects of CME size. Hence, CMEs with large radius ($k >> 1$) or intense bulk deceleration ($\frac{\Delta\dot{r}}{\dot{r}} << 0$) would have stronger radial decelerations. Nevertheless, as we commented on before,

the value of $\epsilon$ may change depending on the effects of the solar wind on the expansion of CMEs.

In the particular case of the expression we are using, Gulisano et al. (2010) obtains values for $\epsilon$ of $0.89\pm0.15$ and $0.45\pm0.16$ for those unperturbed and most perturbed CMEs, respectively. Hence, departing from such a criteria, the expansion rate of those unperturbed CMEs ($\epsilon \sim 1$) would likely depend on $\Delta\dot{r}$, rather than $k$. In contrast, for the cases of perturbed events ($\epsilon < 1$), we expect that CME size ($k$) would dominate over the proportional acceleration of CME center. We illustrate this in the left bottom

panel of Figure 6, where we plot the values of $\Delta\dot{R}$ and Equation (17) (dashed line) as functions of $k$. In the panel we note that





the data follow our semi-empirical tendency, particularly when considering the error bars associated with the effects of $\Delta \dot{r}$. Here, we remark that the relation between CME size and expansion rate deceleration was previously reported by Démoulin et al. (2008).

In addition, although the value of $\Delta \dot{r}$ is in general low, for completeness purposes we explored for the main conditions that may drive the value of bulk deceleration. We found that the relative speed between CME leading edge and solar wind ahead the CME is a determinant factor for bulk deceleration; we can appreciate this in the right bottom panel of Figure (6). In the panel we show how the proportional bulk deceleration of CMEs intensifies as the difference $\dot{L}_1 - u_1$ grows larger; we also plot the regression ($2^{nd}$ degree polynomial) for the data dispersion. In the panel we note a tendency for $\Delta \dot{r}$ to decrease as the value of $\dot{L}_1 - u_1$ grows larger; and it seems to vanish when the *in situ* speeds tend to equalize each other. Hence, faster CMEs would have stronger bulk decelerations and, in consequence, more intense expansion rate decelerations. In consequence, as long a CME presents a self-similar-like expansion (*i.e.*, Equation (17)), we would expect that fast CMEs with large radii would have stronger radial decelerations.

### 3.2 Synthetic profiles of magnetic intensity

With an average error of $4.5\,nT$ (see Table 2), our calculations of magnetic intensity had the second best performance between synthetic profiles. In Figures 2, 4 and 5 we note that our results (blue solid lines) qualitatively resemble the *in situ* data they are attempting to approximate, with most of the proportional errors in the range of 30% and 15% (see yellow bars in Figure 3). However, it is important to remark that we selected the values of the free parameter $b$ that minimized the error ($\varepsilon_B$) in our results; implying that our errors cannot be reduced further.

Although all our synthetic profiles showed the hill-like shape characteristic of the Linquist solution, we found three effects that may modify the way a synthetic profile is observed: *i)* the decrease in magnetic intensity due to the expansion of CMEs ($\Delta \dot{R}$); *ii)* the assymetry driven by deceleration of expansion rates; and *iii)* the path at which the magnetic field is 'measured' (*seen*) inside CMEs, *i.e.*, the impact parameter. The effects of CME expansion on magnetic field are well known, as well the consequences of the impact parameter on the measured data. However, the effect of $\Delta \dot{R}$ is not commonly explored; to the best of our knowledge, only Démoulin et al. (2008) has discussed this topic.

The Figure 7 illustrates the effects of $\Delta \dot{R}$, and impact parameter, on the observed magnetic field symmetry. The left upper and bottom panels of the Figure show the synthetic profiles of magnetic field for the events with the strongest and weakest expansion rate decelerations, respectively. If we focus on the solid bold profile ($0°$) in the bottom panel, we appreciate a significant symmetry that makes the peak (open square) of magnetic intensity to appear near ($\sim 6\,h$) the midpoint (open triangle) of $\Delta T$ ($\sim 16\,h$). Conversely, in the upper panel the peak of magnetic intensity occurs early during $\Delta T$, even before the transit of the CME center (open diamond), a condition that leads to an accentuated asymmetry. Such an asymmetry is due to a process similar to the one already described in Section (3.1) for speeds, since most of the transit time is spent in the transit of the backside magnetic field, forcing the leading magnetic field to rapidly transit by the 'spacecraft'.

In the (a) and (c) panels of Figure 7 we also show the magnetic profiles computed for a number of angular separations between the 'measurement location' and CME axis, that run from the CME center ($0°$) to the CME boundary edge. It is





345    important to comment that such an angular separation (impact parameter) relies on the CME size, the reason being that Event
       7 ($k = 2.93$) has larger angular separations than Event 5 ($k = 0.88$). We note in both panels, that the hill-like shapes gradually
       flatten out, and the overall intensity decreases, as the angular separation between CME and measurement location grows larger.
       Unexpectedly, this flattening also reduces the asymmetry in the profiles of panel (a), which starts as an accentuated asymmetric
       profile ($0°$) to end as a constant-slope-like trace of short duration ($20°$). Whereas the symmetry in profiles of panel (c) is barely
perturbed by the angular separation, and we also observe the already commented reduction in transit times. We remark that the
       profiles in the left upper panel of Figure 8 have similar properties to those of the three groups defined by Jian et al. (2006),
       which used the total perpendicular pressure as a proxy to define the trajectory inside a CME-like structure.

       Hence, according our formalism, the asymmetry of magnetic intensity profiles is closely related with $\Delta\dot{R}$, as we illustrate
       in the panel (b) of Figure 7. The panel shows the calculated moment for the transit of magnetic intensity peaks, normalized by
$\Delta T$, as function of $\Delta\dot{R}$. We note in the panel that magnetic peaks appear early for strong decelerations; and, as the deceleration
       decreases, the appearance of magnetic peaks tends to delay. Furthermore, when $\Delta\dot{R} \sim 0$ the transit of magnetic peaks is closed
       to $\Delta T/2$, as the data regression suggests (dashed line). As a consequence, due to the symmetry of magnetic profiles being
       mainly an effect of $\Delta\dot{R}$, we expect that larger and faster CMEs would tend to have asymmetric-like magnetic intensity profiles,
       than those slow and small ones; again in agreement with the results of Démoulin et al. (2008).

In addition, we found that synthetic profiles systematically underestimated the early *in situ* values of magnetic intensity
       of CMEs. This is particularly clear for events 2, 7, and 9, for which the *in situ* data is larger than the synthetic transits. It
       is important to highlight that those events also had the three largest proportional errors for magnetic field (see Figure 3).
       We believe that such an underestimation derives from a compression by the solar wind that pushes back the frontal regions
       of CMEs in order to decelerate them, processes that simultaneously drives a geometrical deformation and an increment on
magnetic intensity. In Figure 7 (d) we note that the absolute error for magnetic intensity (open diamonds) tends to grow
       larger as the initial speed of CMEs ($\dot{L}_0$) increases. Such a tendency (dashed line) suggests on one hand, that our formalism
       capability to approximate magnetic intensity profiles relies on the initial conditions of CMEs; where faster CMEs would have
       larger associated uncertainties. On the other hand, due to the dependence on the initial speed of CMEs, it would be likely
       that the hypothetical compression on magnetic field would occur during the early stages of CMEs evolution, rather than their
interplanetary propagation.

       Although the behaviour noted above cannot be addressed by our formalism; there are attempts to theoretically solve these
       kind of magnetic profiles. For example, Romashets and Vandas (2005) addressed those profiles via asymmetric magnetic fields
       expressed as an expansion of Bessel's functions. Another example was performed by Vandas et al. (2005), who explored the
       effects on magnetic profiles when an oblate shape is assumed for the flux rope.

### 3.3   CME center, transit times and travel times

       The certainest on the trajectory of a CME as a whole may help to know the transit of the CME boundaries as well as the
       closest approach to the CME center. Although it might be intuitive to relate peaks of magnetic intensity, or related quantities,
       with the magnetic core of CMEs (*e.g.* Jian et al., 2006), those peaks, however, not necessarily approximate the moment of





closest approach to the CME center. The panels (a) and (c) of Figure 7 compare the peaks of magnetic intensity with the

calculated closest approaches to the CME center (open diamonds) for a number of impact parameters for events 7 (asymmetric profiles) and 5 (symmetric profiles), respectively. In the case of event 7 (panel a) we note the substantial differences between the magnetic peaks and the calculated transits for CME center ($TT_r$). Conversely, we note in panel (c) that peaks of magnetic intensity are systematically close ($\sim 1\,h$) to $TT_r$ (symmetric profiles); suggesting that peaks of magnetic intensity are good proxies for CME center transits in symmetric profiles only.

It could also be reasonable to assume the midpoint of transit time ($\Delta T/2$) as an approximation for $TT_r$; values that we also plotted in panels (a) and (c) of Figure **??** as open triangles. We appreciate in the figure's panel (c) that $TT_r$ is near ($< 1\,h$) to $\Delta T/2$; whereas in the panel (a) we note that the CME center and midpoint transit times significantly differ each other. Thus, as it was the case for magnetic peaks, $\Delta T/2$ would approximate $TT_r$ solely for those symmetric profiles of magnetic intensity, *i.e.* for small and slow CMEs. Nevertheless, we highlight that $TT_r$ systematically falls in between the magnetic peaks and transit

time midpoints, regardless of $\dot{R}$ value or the impact parameter either. Hence, in principle, it might be possible to approximate $TT_r$ as the average of $\Delta T/2$ and the occurrence of the peak of magnetic intensity for both, symmetric and asymmetric profiles.

Another method to estimate the closest approach to CME center is the accumulated magnetic flux (AMF). The AMF method uses the maximum variance technique to infer the reference frame of the CME magnetic field, and uses the magnetic coordinate of largest variance to calculate the accumulated magnetic flux. Once the accumulated flux is known as function of time, this

method associates the extreme value of the AMF extreme value to the CME center's closest approach (see Dasso et al., 2006, and references therein for further details). In Figures 2, 4, and 5 we plot the calculated AMF as thin gray lines in magnetic intensity panels for all events. Additionally, we mark out the AMF's extreme values by open gray diamonds; values that we compared with our calculated $TT_r$.

Our method showed a quantitative capability to approximate CME center transits estimated by the AMF method with an

average error of $\sim 9\%$ (see Figure 3). Additionally, Table 2 shows the absolute errors ($\varepsilon_{TT_r}$) associated with our results; we note that, on average, our results differ by a few hours ($\sim 6\,h$) from those calculated by AMF. We highlight that such an error is small when compared with the averages of $TT$ ($\sim 47\,h$) and $\Delta T$ ($\sim 38\,h$). The consistency between the data and our results can be appreciated in Figures 2, 4, and 5 in which the our calculated $TT_r$ (vertical dashed red lines) are systematically close to the extremes of magnetic flux (open diamonds).

It is important to comment that the AMF method assumes a trajectory near a single magnetic structure inside CMEs. Then, large impact parameters or imprecise CME boundaries might mislead the method's results; as well CMEs of non-single magnetic structure. Perhaps one of them is the reason for the errors above the average ($\varepsilon_{TT_r} >> 9\%$) of events 1, 2, 9, and 10. By inspecting these events we notice that their temperature profiles surpassed the expected temperature (solid gray line), a condition that could have a number of explanations. For example, it is reasonable to think of multiple magnetic structures forming

the CMEs, and to assume that the CME material might have been somehow externally compressed. It might also be possible that the CME boundaries are ambiguously determined, implying that we are not correctly analysing the CMEs.

In regard to transit and travel times and impact parameter, we note in Figure 8 that $\Delta T$ is highly dependent on the trajectory at which it is 'measured'(calculated). In both panels we appreciate that $\Delta T$ is maximum when the measuring location passes





by the CME center (0° lines), since the whole and expanding CME is transited. After this maximum, $\Delta T$ gradually decreases
as the CME trajectory moves away from the measurement location (larger angles), which reduces the CME structure 'seen' at
the measured location; implying that the CME radius 'seen' *in situ*, *i.e.*, the value of $k$, would be a lower limit. In contrast, we
appreciate that $TT$ grows larger as the impact parameter gets larger. Of course this derives from the fact that CME structure
delays in being 'seen' the measured point moves away from the CME trajectory. Surprisingly, the growth of $TT$ and the
shortening of $\Delta T$ somehow equilibrate with each other to make the closest approach of the CME center (open diamonds)
almost equal for all impact parameters.

### 3.4   Density and temperature errors

The synthetic profiles of temperature and density were the ones with the largest errors, with averages of 83% the first and
147% the latter (see Figure 3). In particular, our density profiles systematically had errors above 70% that reach values as large
as 327%. Such large errors represent an important limitation for our formalism. Consequently, before we discuss our results
regarding temperature and density, we attempt to understand these errors. In order to do so, we depart from the fact that *in situ*
values of density and temperature showed significant (large and fast) variations during $\Delta T$. This behaviour can be appreciated
in Figure 8 (a), where we present a histogram of the standard deviations ($\sigma$), in terms of the median values of $N_p$ (orange) and
$T_p$ (green), during $\Delta T$. In the panel we note that, when neglecting events 5 and 6, the values of $\sigma$ are systematically larger
than their associated median values ($\sigma > 1$). In the case of events 5 and 6, we had that their median values were far above the
average; nevertheless their standard deviations stayed near the averages; conditions that derived into the short bars shown in
the histogram.

   Therefore, the large variations in temperature and density overwhelm (or mask) their "own" values, an effect that is accen-
tuated in density, with $\sigma > 2 < N_p >$ for five events. Hence, this "masking" effect could be a reasonable source for the large
errors associated with synthetic profiles of density and temperature, as well. In order to explore that, we present the errors ($\varepsilon$)
for density and temperature in terms of their associated $\sigma$ in Figure 8 (b). In the panel we note that all temperature errors (green
bars) are less than their standard deviation ($\varepsilon_T < \sigma_T$), confirming that temperature variations are larger than our error. We also
note a similar behaviour for density (orange bars), where most of errors are less than the variations of data ($\varepsilon_{Np} < \sigma_{Np}$). In
the Figure 8 (b) we also plot the errors calculated for median values of density (gray bars), which are significantly less than
($\varepsilon_{<N>} < \varepsilon_{Np}$); with the exception of event 5 and 10 where $\varepsilon_{<N>} > \varepsilon_{Np}$. We believe this decrease in error between $\varepsilon_{<N>}$ and
$\varepsilon_{Np}$ is because using median values, instead of the collection of data points, reduces the masking effect, if present.

   As we commented in last paragraph, events 5 and 10 had errors significantly larger than the proper variations of density
data. We interpret this condition as another possible source for error present in these events. In order to identify such an error
source, we searched for conditions that these events shared in common. After inspecting the *in situ* profiles (see Figure 5), we
realized that the events have solar wind behind (yelow solid lines) faster than the solar wind ahead of them (green solid lines);
such solar wind being even faster than the CME tailing regions. Those differences in speeds could be driving a compression
of the CME by the ambient solar wind, a compression that might be the additional source of error commented earlier. Table 1
lists the values for solar wind measurements ahead and behind CMEs.





From a simplified perspective, this implies that events 5 and 10 were undergoing a compression process due to slow and fast solar wind parcels ahead and behind them, respectively. Since, on one hand, the slow solar wind acts as an obstacle to the CMEs propagation, which drives stagnation on the leading material and an increase in the intensity of magnetic field. On the other hand, the fast solar wind pushes events from behind, accelerating and compressing the trailing material of CMEs. Subsequently, we proceed to search the signatures for the compression process in the rest of the events. We found that events 4, 6, and 8 seem to be possibly trapped in between two parcels of slow (ahead) and fast (behind) solar wind.

If the compression process is a source of error, the error must be somehow related with it. Figure 8 (c) and (d) compare $\varepsilon_N$ (left) and $\varepsilon_{<N>}$ (right) as functions of the quotient of behind ($p_2$) and ahead ($p_1$) ram pressures of solar wind (see Table 1). Here we use the quotient of ram pressures as an estimation for compression acting on CMEs, where values near or larger than the unit ($p_2 > p_1$) may indicate an undergoing compression. In the panels we note that both errors tend to grow as the pressures quotient increases, a tendency that seems to be linear (solid black lines). We note in panel (c) that, when the quotient tends to vanish, $\varepsilon_{Np}/\sigma$ converges to a value around $\sim 0.4$. On the other hand, in the case of $\varepsilon_{<N>}$ (panel d), the error tends to vanish when the pressures quotient approaches to zero. We interpret the residual error in the case of $\varepsilon_{Np}$ (panel c) as a general value for the masking effect, since it seems to vanish in the case of $\varepsilon_{<N>}$ (panel d).

Earlier, we isolated two possible error sources for our results. First, we had the *masking effect* related with an intrinsic property of the data used for our analysis. Second, we had the effects of compression, that derives from the conditions at which an event evolves. Although the effects of the first source of error could be reduced by comparing median instead of instantaneous values; we were unable to remove, nor to reduce, the effects of compression in our errors. Nevertheless, the quotient of ram pressures seems to be useful to determine the magnitude of error that compression would have on our results. This is particularly important, since an external compression may modify the bulk speed, density, temperature, and the magnetic field magnitude of CMEs. Additionally, if the CMEs are undergoing a compression at their boundaries, it should also affect the CME's shape, turning our circular cross section into a pancake-like one (see Hidalgo et al., 2002; Hidalgo, 2003; **?**; Riley et al., 2004; Nieves-Chinchilla et al., 2005). All those modifications clearly deviate our model's results from the real case. However, we remark that the large errors derive from the inherent complexity of the phenomena we are studying, a complexity that our model is unable to reproduce in detail. Thus, with the possible sources of errors already identified, we proceed to discuss our density and temperature results.

### 3.5 Synthetic profiles of density and temperature

The calculated profiles of density (solid blue lines in density panel) in Figures 2, 4, and 5 show a rarefying tendency during $\Delta T$ commonly associated with the aging effect of CMEs. Note that logarithmic scales in vertical axis might obstruct the detection of such a tendency. This rarefaction, in general, is also present in *in situ* data, for which the CME density usually starts with values of $\sim 4\,\mathrm{cm}^{-3}$, and ends with significant less values. This decrease in density is commonly thought to be provoked by CME expansion. In our approach, this rarefaction process is also driven by an expansion, since $N_p$ is inversely proportional to the $\theta r R^2$ product (see Equation (13)).



As we already commented on before, the density profiles showed large proportional errors (see Figure 3). However, when we compare those errors with their corresponding $\sigma$ (right panel of Figure 8), only in four cases were they of significance ($\varepsilon_{Np} > \sigma$). Furthermore, the errors of the median values ($\varepsilon_{<N>}$) significantly decreased, except for those events under strong compression. Surprisingly, when neglecting those potentially compressed events (5, 8, and 10), the absolute value of $\varepsilon_{<N>}$ fell from 101.5% to 45.6% (see Figure 3).

In the case of temperatures, we note that our calculated profiles do not seem to be affected by the CME expansion, since they barely change during $\Delta T$. This apparent behaviour is due to the near-unit value for $\gamma$, which makes the exponent of Equation (14) to be zero. This apparently-constant tendency is not clear in the *in-situ* data, perhaps only event 3 shows it, and events 1 and 5 resemble such a tendency.

Although the median of synthetic temperature profiles equal their *in situ* data counterparts by construction, the proportional values of $\varepsilon_T$ are large, with an average value of 60%, as Figure 3 shows. Although the errors in temperature may seem large, we remark that they are less than the proper variations found in temperature data during the CME *in situ* transits. Because, for all cases, $\varepsilon_T < \sigma_T$ with an average of $\sim 0.6\sigma$ (see right panel of Figure 8). In contrast with density, temperature seems not to be affected by compression; since events 5 and 10 did not have errors larger than the average. This could be caused by the near-zero value of the exponent in Equation (14), which would make temperature almost unaffected by changes in density (or pressure).

The two potential sources of error we described may cause the large inconsistencies in the synthetic density profiles. On one hand, if masking and compression effects are actually playing roles in CME evolution, it would mean that some of our assumptions may be partially satisfied. For example, the assumption of isolated events, or mass homogeneously distributed through the CME volume, and thermodynamic equilibrium would be not fulfilled, at least, for two events. On the other hand, the masking effect ($\sigma_{Np} > < N_p >$) would lead to significant large errors when comparing a collection of data points. The large errors in density, and their possible sources, reveal some limitations in our approach, which cannot reproduce the complexity in density and temperature found inside the events analyzed. Nevertheless, our modeling may offer a simplified glimpse concerning the general evolution of CMEs as a whole.

## 4 Summary and Discussion

In this work we presented a formalism to compute *in situ* transits of fast (super-magnetosonic) Earth-directed CMEs. Our model consists of a collection of simple relations to calculate synthetic profiles of *in situ* measurements as would be seen during the transit of fast CMEs across the Earth's orbit. The synthetic profiles our model calculates are: the radial component of speed (Equation (12)), density (Equation (13)) and temperature (Equation (14)) of protons, and magnetic magnitude (Equation (15)). The travel time of CME center (Equation (9)) and total mass of CMEs (Equation (10)) can be approximated as well.

Our formalism combines analytic models and empirical tendencies, conditions that allow us to keep it simple and easy to implement, as compared to MHD approaches. We assumed the geometry of CMEs as cylinders of circular cross section whose radius is given by the self-similar empirical relation found by Bothmer and Schwenn (1998) and later verified by Gulisano et al.





(2010). The trajectories of CME leading edges were calculated with the 'piston-shock' model (Corona-Romero et al., 2013,
2015), which assumes an isolated and fast CME propagating through an almost-quiet ambient solar wind. We approximated
the magnetic field inside CMEs by the well known Lundquist (1951) solution, whose intensity decayed due to the radial and
longitudinal expansion of CMEs; decaying that followed the empirical tendency by Gulisano et al. (2010). In addition, to solve
the density and temperature of protons inside CMEs, we assumed the CME's material to be a polytropic plasma in thermal
equilibrium and homogeneously distributed within CMEs.

Our approach has some obvious practical benefits. Unlike global MHD models, which require significant time both in
development of the algorithms, running of the codes, and time spent analyzing and visualizing the results, our technique is
simple to implement and interpret. Additionally, it requires extremely modest computational resources; and the results can be
compared directly against *in-situ* measurements for specific events, providing direct feedback for the quality of fit, and, hence,
the likely accuracy of the solution. Besides, our formalism's simplicity may also provide unique insight to the dynamical
processes at work as the CME propagates away from the Sun. Although they are included in the more sophisticated numerical
approaches, the complexity of them often masks the underlying mechanisms. Furthermore, we explicitly separate the CME's
propagation into a short interval of constant speed followed by a period during which the CME asymptotically approaches the
speed of the solar wind, which may represent distinct underlying phases in the CMEs evolution.

This simplicity also comes with limitations, mainly associated with our physical assumptions. Perhaps the more evident
examples are those related with density and temperature errors, where the hypothesis of homogeneously distributed matter and
thermal equilibrium contrast with the *in situ* data that showed rapid variations and complex profiles. Such a behaviour could be
signature of inner structure inside CMEs like multiple flux ropes (Hu et al., 2004; van Driel-Gesztelyi et al., 2008), as might be
the case of event 7 (see Dasso et al., 2009), or even processes at the interior of CMEs, like internal shocks (Lugaz et al., 2015).
For construction, our formalism neglects inner structures, and processes, inside CMEs. For this reason, our synthetic profiles
cannot reproduce the complexity of observed *in situ* data.

Also related with the inner structure of CMEs, the magnetic field used in our approach is only suitable for a single flux rope
and it is unable to be adapted for more complex scenarios like multiple flux ropes or oblate shapes of CMEs. Additionally,
our fixed geometry obstructs our formalism to include the effects of, for example, the pressure due to surrounding solar wind;
which we found to be of significance for some events. These magnetic and geometrical conditions make our formalism more
suitable for the core of flux rope CMEs than the whole CME structure. Hence, for complex scenarios, our model's simplicity
becomes a weakness.

There are alternatives, if not to address, at least to reduce the effects of some of our model's limitations. In the case of oblate
or 'pancake' shapes provoked by an asymmetric expansion of CMEs, we could use an elliptical cross section instead of a
circular one (*e.g.* Vandas and Romashets, 2017a). For this case, the eccentricity could be taken constant or might somehow be
estimated by the pressure on CME by the surrounding solar wind. This geometrical change, however, would not significantly
affect the trajectory neither the descriptions of density and temperature. Conversely, the Linquist solution would not longer be
valid for this scenario, and the magnetic field would require a more sophisticated solution for generalized geometries like those
proposed by Vandas and Romashets (2003); Owens et al. (2012), and, Vandas and Romashets (2017b) among of others.





Other limitations for our formalism come from the quiet ambient wind and isolated CMEs hypothesis that are requirements
of the analytic model used to approximate the trajectories of CMEs. In the case of the ambient solar wind, experience dictates
that it is unlikely to observe quiet solar wind for large periods of time, even during the solar minimum, when there are multiple
interacting regions due to coronal holes dispersed all over the solar disk. In addition, during, or near the solar maximum, high
solar activity rates may break the isolation assumption. Corona-Romero et al. (2017) also found those limitations and managed
them as uncertainties associated to the results computed by the piston-shock model. In such a context, the uncertainty would
be represented by upper and lower limits for the possible synthetic profiles.

Despite our assumptions on geometry, density and, temperature might seem restrictive; they are in agreement with previous
empirical results. Since we assumed CME mass to be constant, the change in $N_p$ is defined only by the expansion of CMEs,
*i.e.* the volume changes. In our approach, $N_p$ decreases as $r^{-(2\epsilon+1)} = r^{-2.56}$ (see Equation (13)), which is in agreement with
the empirical estimations found by Bothmer and Schwenn (1998) ($N_p \propto r^{-2.4\pm0.3}$) and Liu et al. (2005) ($N_p \propto r^{-2.32\pm0.07}$).
In the case of $T_p$, Liu et al. (2005) found that $T_p \propto r^{-0.32\pm0.06}$, a result surprisingly similar to the one deduced in this work:
$r^{-(\gamma-1)(2\epsilon+1)} = r^{-0.36}$ (see Equation (14)). The aforementioned consistencies between our expressions and those empirically
found suggest that our modelling of CME volume and its material approximates sufficiently well the empirical cases.

Perhaps the weakest point in our approach, from a physical perspective, is the expression for magnetic field intensity, for
which we combined the Lundquist solution and the self-similar empirical tendency for the decaying of magnetic intensity
with heliocentric distance. Although it is an straightforward expression to approximate representative data, it keeps similarities
with the theoretical approach described by Démoulin et al. (2008), who applied similar geometrical conditions. In such a
theoretical approach, the magnetic intensity for an isotropic expansion decays as $e^{-2}$, where $e$ is a time-dependent factor that
normalizes the distance from the CME center in the Bessel functions. In our case, such a normalizing factor is proportional
to $R$ and, for consistency, $R^{-2}$ should be similar to the empirical tendency that express the intensity decaying in Equation
(15). We can verify this since $R^2 \propto r^{2\times(0.78\pm0.12)} \sim r^{-1.56\pm0.24}$, whereas the previously described empirical tendency goes as
$\sim r^{-1.85\pm0.07}$, values that are close each other, especially when considering the uncertainties. Hence, although Equation (15)
is an *ad hoc* expression to approximate the magnetic field intensity, it is consistent with the theoretical approach by Démoulin
et al. (2008).

Another simplification we used applied to the orientation of the CME, which we restrict to CMEs whose associated active
regions were near the center of the solar disk. Furthermore, it is precisely those events, with the solar disk center as source
region, that are likely to have the strongest geomagnetic effects. Such a restriction allowed us to assume that the spacecraft
intercepts the CME near its symmetry axis, and kept our expressions as simpler as possible as we aimed in this introductory
work. The only case for which we superficially investigated the effects of deviation from the CME axis on our synthetics
profiles was for the magnetic intensity. Although such an exploration gave a first glimpse about the way magnetic profiles and
travel times are affected by the spacecraft trajectory; the exploration also requires us to contemplate rotation of the CME itself.
It is important to comment that the additional degrees of freedom due to rotation and displacement may help in reduce the error
for magnetic intensity profiles. We reserve as future work a geometrical generalization in which we will solve a more general
approach.





Our synthetic speed profiles showed the decreasing tendency regularly associated with the *aging* effect. The *aging* could
express it self as a constant-slope or a curve-like tendency and is manly driven by the expansion of CMEs. However, we found
that deceleration of expansion rate of CMEs is highly related with the effects if *aging* in such a way that intense(negligible)
decelerations would generate curve(constant-slope)-like speed profiles. Additionally, as long as the CME expansion could be
modelled by a self-similar expression, fast(slow) and large(small) CMEs would have larger(smaller) decelerations in expansion
rates.

In addition, we also found that deceleration of expansion rate of CMEs also affects the symmetry of magnetic fields profiles,
making the magnetic peak to appear earlier than the CME center (see discussion of Figure 7). For this case, the asymmetry
grew larger with the intensification of deceleration, and for the hypothetical case of negligible deceleration (slow and small
CMEs) we would expect highly-symmetrical profiles of magnetic intensity. Finally, we observed that the average between the
peaks of magnetic intensity and the midpoint of transit times were consistent with the travel time of CME centers, conditions
that hold for different trajectories, speeds and sizes.

We realized that compression by solar wind may affect the *in situ* transit profiles of CMEs, consistent with the results
reported by Démoulin and Dasso (2009). For example, we found evidence between the compression by the solar wind and our
error to compute the CME density. Furthermore, it is well known that solar wind effects may affect the geometry of CMEs
and, with it, their inner properties. We believe that such a compression could be the cause for large magnetic intensities at the
frontal regions of CMEs. Additionally, other works also explore such a process that could affect the selfsimilar expansion of
CMEs by modifying the value of $\epsilon$ (*e.g.* Gulisano et al., 2010).

### 4.1 Validation and Results

We validate our formalism by comparing its results with empirical data. Another way to assess the technique described here
would be to compare directly with MHD results. Although there may be approximation and assumptions embedded within
global MHD results, they likely represent a much more accurate approximation to the actual dynamic evolution of CMEs.
Thus, by extracting a set of solar and interplanetary pseudo measurements from a selection of MHD results, we can test
our approach in a more controlled scenario, where the actual inputs and outputs are exactly known. This kind of numerical
experiment was used to test a variety of force-free flux rope models in the past (*e.g.* Riley et al., 2004). Such an approach will
be useful when extending our approach for the general case of CMEs not aligned with the Sun-Earth line of sight.

In the Section 3 we computed and analysed the synthetic profiles of speed, density, temperature, and magnetic intensity for
10 fast (Earth-directed) halo CMEs detected during the period 2000-20015 (see Table 1). In order to do the calculations we used
physical data from the events analysed, and free parameters whose values were carefully selected (see Section 4.2 for further
details). Our results indicated that synthetic profiles of speed had the best performance, followed by the magnetic intensity
ones, with average errors of 9.6% and 27.6%, respectively. In contrast, temperature and density of protons had larger errors,
615   with averages of 83% for temperature and 46% for density, when neglecting the potentially compressed events. Additionally,
the travel times of CME center, which we also calculated, had an average error of 9%.





Regarding of speed profiles, we remark that they closely followed their *in-situ* registered counterparts, with proportional and absolute errors below 17% and 120 km s$^{-1}$, respectively. Our speed profiles depend on the values of bulk (CME center) speeds and expansion rates (radial speeds) of CMEs, speeds that had decelerations of 8% and 19% as average, respectively. Hence, our results suggest that, in average, the bulk speed of CMEs barely decelerates during the transit though out the Earth's orbit, whereas the deceleration of expansion rate is still significant. Those decelerations are of interest since models of magnetic field commonly assume them as negligible, assumption that contrast with our results.

Our synthetic profiles of magnetic intensity qualitatively approximate their associated *in situ* values with absolute errors within the range of 2.11 nT to 9.14 nT, and an average of 4.53 nT. We noted that for those events with larger initial speeds, our synthetic profiles underestimated the early values of their *in situ* registered counterparts. Such underestimation generated large errors in such events, and it is likely due to a compression of solar wind on the frontal region of CMEs during the early stages of their propagation. Furthermore, all our synthetic profiles showed the characteristic hill-like (bell-like) shape of the Linquist solution for flux ropes, a shape that was significantly influenced by the *aging*, as we noted above.

The synthetic profiles of density had the largest errors, which potentially had two sources: *i)* a masking effect due to the large and fast variations in *in situ* data; and *ii)* a compression of the CME material due to the ambient solar wind. We managed to reduce the effect of large variations by calculating and comparing median values instead of instantaneous ones; procedure that made the averaged error of seven (not compressed) events to fall from 112% to 46%. Such a behaviour contrast with those events overtaken (compressed) by fast solar wind, whose errors barely changed after the median-value treatment. We realized that the error in density is directly related to the quotient of solar wind ram pressure in such a way that, when the solar wind compression is negligible, it seems that the masking effect is the main source for error for our density results (see Figure 8).

Our synthetic density profiles reproduced the expected decreasing-with-time tendency due to the CME expansion, and approximated the measured median values for many cases as well. We remark that synthetic density is the only profile that does not have a free parameter directly associated with it. Nevertheless, it is highly sensitive to the ion content (*e.g.*, alpha particles) and the angular width of CMEs, which we assumed constant. Those physical properties could be used as free parameters whose values we could also select to decrease the error associated with our density results. As an example, by changing the value of $q_\alpha$ in Equation (13) from 12% to 5% or 20%, we would induce variations on density of $+23\%$ and $-18\%$, respectively.

Regarding the synthetic profiles of temperature, we used a free parameter to match the median value of our synthetic profiles with their *in situ* median counterparts (see description of Equation (14)). Therefore, for construction, our synthetic profiles were representative of the data they aimed to approximate; nevertheless, the averaged absolute error was of $41 \times 10^3$ K ($\sim 83\%$). Such a large error was mainly a result of by the fast and accentuated variations in *in situ* data (*i.e.* masking effect) that, in some cases, was larger than. or of the same order in magnitude (see Events 1, 2, 7, and 10). In fact, most events showed a complex temperature profile that even surpassed the *expected temperature* of protons; a condition that might suggest anomalous structures within the CMEs like multiple magnetic structures, or processes with the capacity to modify the inner structure of CMEs, like internal shocks, or even the afore mentioned compression by the surrounding solar wind.

Besides synthetic transit profiles, we also calculated the time spent by the CME center in traveling from near the Sun up to the Earth's orbit, *i.e.*, travel times of the CME center. We compared our results with those obtained by the accumulative magnetic





flux (AMF) method. Our results quantitatively approximated the *in situ* transits estimated by the AMF method with an average error of $\sim 9\%$ corresponding to $\sim 6\,h$. Surprisingly, according to our results, the travel times of the closest approached point to the CME center were not significantly affected by the trajectory between CMEs and the point measurement.

## 4.2 Free parameters

To compute synthetic transits our formalism uses four free parameters related with the inertia ($c$), geometry ($k$), temperature ($T^*$), and magnetic intensity ($b$) of CMEs. The adequate selection of free parameters allowed us to approximate the *in situ* transit profiles of fast CMEs, selection that followed the next criteria: The CME inertia ($c$) was estimated by forcing the piston-shock model to match the calculated travel times with their *in situ* measured counterparts. The radius of each CME ($k$) was estimated through the *in situ* measured transit times of CMEs. The temperature of CMEs ($T^*$) was set by requiring our calculated median of temperature be equal to its *in situ* counterpart. Finally, the value of $b$ was selected to minimize the absolute difference between the synthetic magnetic profile and its *in situ* counterpart. In our formalism, as we already commented, there are potentially other additional free parameters, such as the semi-angular width of CMEs ($\theta$) and CME's alpha particle content ($q_\alpha$), whose values were assumed constant and equal for all the events we analysed.

Previous works (Corona-Romero et al., 2017) have shown that the free parameter $c$ can be approximated by:

$$c\left[\frac{\dot{L}_0 - u_1}{1\,\mathrm{km\,s^{-1}}}\right] = 3380.6\left[\frac{\tau_f}{1\,\mathrm{h}}\right]^{-1.14}. \tag{18}$$

This expression that relates the CME inertia with rise phase duration of solar flare, and solar wind speed. We plot Equation (18) in the Figure 9 (a). The panel also shows the data used in our present analysis (open diamonds) for comparison purposes. Based on the results of Corona-Romero et al. (2017), we searched for possible relations between our other three free parameters and the input data. In order to do so, we performed a parametric study that led us to find three tentative relations.

We show the results from our parametric exploration in the panels of Figure 9. In the panel (b) we plot (open diamonds) the product $\tau_f\,k$ as function of the proton density of solar wind ($n_1$), which can be approximated by:

$$k\left[\frac{\tau_f}{1\,\mathrm{h}}\right] = 58.4\left[\frac{n_1}{1\,\mathrm{cm^{-3}}}\right]^{-4.9} + 0.9. \tag{19}$$

This is also plotted in Figure 9 (b) as a solid line. According to Equation (19), for a given $\tau_f$, the CME radius is somehow inversely related with the solar wind density up to a limit value, after which, it stays constant. This could be due to effects of solar wind inertia on the expansion of CMEs, since larger inertias (densities) of solar wind would evolve into slower expansion rates, which should lead to shorter radii of CMEs, consistent with Démoulin and Dasso (2009).

Concerning the temperature, we found that the value of $T^*$ could be approximated by:

$$T^* = 1.5\left[\frac{P_{ram}}{1\,\mathrm{nPa}}\right] + 0.366, \tag{20}$$

with $P_{ram} = m_p(1 + p_2/p_1)\,p_1$ the sum of ahead ($p_1$) and behind ($p_2$) ram pressure ($p = nu^2/2$) of solar wind (see solid line in left-bottom panel of Figure 9). Equation (20) suggests that the more(less) compressed by the surrounding solar wind a CME





is; the hotter(colder) than the average it would be. It is important to note that, when the effects of solar wind compression are negligible ($p_2 < p_1$), the value of $T^*$ could be satisfactorily approximated by solar wind ($p_1$) ahead solely. Whereas, in the case of magnetic field intensity, we found that $b^2$ seems to be related with $\tau_f^2/(\dot{L}_0 - u_1)$ according to:

$$685 \quad b^2 = 6.72 \times 10^{-11} \left( \left[ \frac{\dot{L}_0 - u_1}{1\,\mathrm{km\,s^{-1}}} \right]^2 \left[ \frac{\tau_f}{1\,\mathrm{h}} \right]^{-2} \right)^{1.91} + 426.8\,. \qquad (21)$$

This relations is also plotted (solid line) in the panel. Thus, according Equation (21), we would expect stronger magnetic fields within fast CMEs related with solar flares of short duration. This relationship appears to have a lower limit, when a slow CME is associated with a prolonged solar flare.

By combining Equation (18) and the piston-shock model, Corona-Romero et al. (2017) were able to forecast travel times, arrival speeds, and even to estimate masses of fast CMEs. Thus, it is reasonable to infer that Equations (19) to (21) could be a possible way to approximate the expected values of free parameters. In such a hypothetical case, those relationships would help to specify all the required data to calculate synthetic transits before an event impacts the Earth; *i.e.* we could perform analytic forecasting of *in-situ* transits of CMEs. Since once an event is identified by coronagraph, it would be possible to collect the values of $L_0$, $\dot{L}_0$, $\tau_f$, $n_1$, and $u_1$, we would proceed to calculate the values of $c$, $k$, and $b$; and, by neglecting the effects of compression, we would approximate the value of $T^*$. After, with all those values known, we would apply our formalism to compute the *in situ* transit of the event as possibly would be seen at Earth's orbit.

To illustrate this, we 'forecast' the synthetic profiles of Event 1. Figure 10 compares the results from this 'forecasting' test (red profiles) and the already calculated synthetic transit of Event 1 (blue profiles). In general terms, we note in the figure that both profiles are close each other, with the forecast profiles approximating their calculated counterparts. Additionally, we also note differences. For example, the 'forecast' CME center (dashed red vertical lines) arrives a few hour earlier than the computed one, and red profile of density is slightly above of its blue counterpart; behaviours that suggest an excess of inertia. Regarding temperature and magnetic intensity forecast profiles, the main change we note is a translation caused by the early arrival noted above. Additionally, the 'forecast' transit time (dotted red vertical lines) is significantly less than its calculated counterpart, which may imply faster bulk speed and, or, a shorter radius. Here, it is important to highlight that the differences between the 'forecast' and Table 2 values of the free parameters were 46.8%, -8.6%, 0.6%, 6.4% for $c$, $k$, $T^*$, and $b$, respectively. Values that explain the early arrival, the excess of density, the shorter transit time, as well as the similitude between temperatures and magnetic field intensities seen in the forecast profile.

Our formalism's forecasting capabilities tentatively rely on Equations (18) to (21). Corona-Romero et al. (2017) recently validated Equation (18). If future studies validate the other remaining three, it would allow us to apply our formalism to systematic forecasting of CME arrivals. In addition, because our speed and magnetic profiles have with lower errors, our work might also help to forecast CME geoeffectiveness, since the product $v^x \times B^z$ plays an important role for this purpose (see Richardson and Cane, 2011, and references therein). In such a case, our forecasting capabilities could be strengthened by combining our formalism with, for example, the approach of (Savani et al., 2015, 2017), which aims to forecast the magnetic polarity of flux ropes and their orientations. Furthermore, by combining our results with previous works Corona-Romero and





Gonzalez-Esparza (*e.g.* 2016), we would be able to simultaneously forecast *in-situ* transit profiles of CMEs and associated shocks/sheaths. The capability to simultaneously forecast *in situ* transits of CMEs, the geoeffectiveness, associated forward shocks, and plasma sheaths is of great interest for space weather purposes, since more intense geomagnetic storms are triggered by such phenomena (Ontiveros and Gonzalez-Esparza, 2010, and references therein). If this formalism is shown to be robust under a range of conditions, it can lead to an important operational tool for space weather, particularly for those scenarios when

the response time is of importance, like early warning systems. Exploring its robustness would be our immediate task.

## 5   Conclusions

We presented a semi-empiric formalism to compute synthetic *in situ* transits of fast Earth-directed halo-CMEs. Our formalism combines analytical and empirical models to develop a method based on simple equations that allows us to approximate the radial speed, density and temperature of protons, and magnetic field intensity during the transit of CMEs as seen at the orbit of

Earth. Although our we compute synthetic transits for an astronomical unit, our equations can be adapted to another heliocentric distances like Mars or elsewhere. Additionally, our formalism also calculates travel time of the CME center and its arrival speed as well.

To compute synthetic *in situ* transits of CMEs, we used data related with the event to analyse. The data our method requires are: *i)* the initial position and speed of CMEs (from coronagraph images), *ii)* the rise phase duration from the associated solar

flare (taken from X-ray fluxes), and *iii)* the solar wind conditions from *in situ* measurements. Whereas, the free parameters were associated with inertia, size, temperature and magnetic field of CMEs, and we used them to tune our results.

We used our formalism to approximate 10 *in situ* transits of fast CMEs occurred during 2000-2015. In this test we found that profiles of speed, magnetic intensity, and temperature had average errors of 10%, 27%, and 83%, respectively. Additionally, the error for the travel time of CME center was of 9%. In the case of density, our results were strongly affected by the solar

wind compression on CMEs, which caused discrepancies with the observations. In this sense, the average error of density for all events was 102%; whereas, neglecting the three events significantly perturbed by the compression effects, the average error dropped to 46%. It is important to remark that errors of temperature and density, even in those compressed cases, were lower than the rapid and large variations inherent in the *in situ* data.

In addition to computing *in situ* transits, we also found that deceleration of CME expansion rate may play an important role

in the way *in situ* transits are 'seen'. On one hand, stronger decelerations apparently provoke a curved-like profiles in speed synthetic transits, this contrast with an constant-slope profiles of CMEs with almost constant expansion rate. On the other hand, we noted that our calculated magnetic-intensity profiles tend to be symmetric(asymmetric) for CMEs with negligible(large) deceleration expansion rates. Surprisingly, those large and fast CMEs would tend to have larger deceleration expansion rates than those smaller and slower ones.

Our formalism relies on a number of assumptions that simplified the conditions in which a fast CME evolves and propagates. From these simplifications arose a number of limitations that may increase the error in our results, particularly for complex events. Nevertheless, our formalism showed to approximate particularly well the speed and magnetic intensity profiles, both

directly related with the geoeffectiveness of CMEs. Besides, we found possible empirical relationships to estimate the free parameters our model requires, which might allow us to implement our method to forecast *in situ* transits of CMEs. Hence, in

conjunction with other approaches, our model can lead to an important operational tool for space weather forecasting, specially in the case of early warning systems.

*Code and data availability.* The data sets and catalogues used in this work are publicly available. Our software is partially available; however, we can help the enthusiastic readers by sharing parts of our code and bringing support for the rest of the coding. Most of our code should be compatible with 'GNU data language' (GDL).

*Author contributions.* P. Corona-Romero: model developer and software writer. P. Riley: reviewer of model's consistency. Both authors analysed and discussed the results, and collaborated in the manuscript's redaction.

*Competing interests.* No competing interests are present.

*Disclaimer.* TEXT

*Acknowledgements.* P. Corona-Romero is grateful for CONACyT Grant 254812. Space Weather Service Mexico (SCiESMEX) is spon-
sored by Catedras-CONACYT Program, project 1045. Pete Riley gratefully acknowledges support from NASA and NOAA (under grants NNX15AF39G and NA18NWS4680081). We acknowledge the Space Physics Data Facility (SPDF) and OMNIWeb for the *in situ* data used in this work. We acknowledge use of NASA/GSFC's Space Physics Data Facility's OMNIWeb service, and OMNI data. We used data from LASCO CME Catalog which is generated and maintained at the CDAW Data Center by NASA and The Catholic University of America in cooperation with the Naval Research Laboratory. SOHO is a project of international cooperation between ESA and NASA.





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



**Figure 5.** Calculated synthetic transit of events 5 to 10. We present two rows with 3 columns each; and each column of 4 panels shows a different event. From top to bottom, the panels present the radial component of the solar wind speed ($|V^x|$), the density ($N_p$) and temperature ($T_p$) of protons and the magnetic field intensity ($B$). Solid blue lines summarize our model results. Dotted and dashed vertical red lines mark the CME boundaries and center, respectively. Dotted red lines mark the CME boundaries. The solid black lines are *in situ* measurements from OMNIWeb service. Green solid lines mark *in situ* solar wind used for calculations (see Table 1). The solid grey lines in $N_p$ and $T_p$ panels are the plasma Beta (10 folded) and the expected proton temperature ($T_{exp}$) (Lopez, 1987), respectively. The solid grey line in $B$ panel is the accumulative magnetic flux, as defined by Dasso et al. (2006), whose extremum (open diamond) gives an estimation for the magnetic center inside the CME.



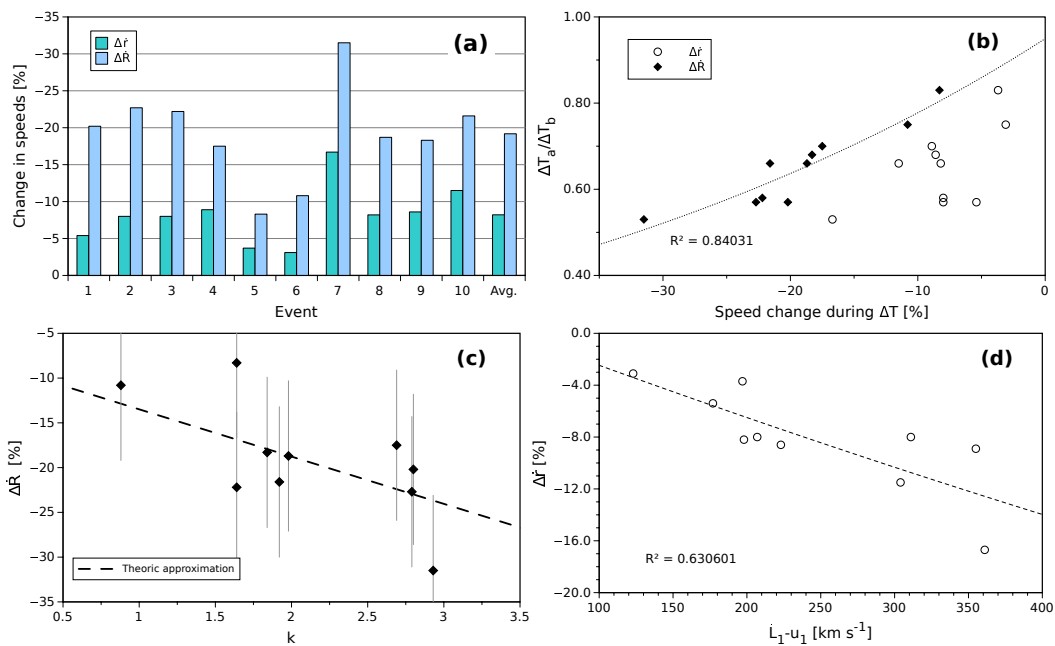

**Figure 6.** The changes on $\dot{R}$ and $\dot{r}$ during $\Delta T$. Panel *(a)*: Histogram of the proportional variations of the CME center $(\Delta \dot{r})$ and expansion $(\Delta \dot{R})$ speeds respect to $\dot{r}(TT_r)$. The right-most columns are the average values of $\Delta \dot{r}$ and $\Delta \dot{R}$, respectively. Panel *(b)*: Data dispersion of $\Delta T_a / \Delta T_b$ as function of $\Delta \dot{R}$ (black diamonds) and $\Delta \dot{r}$ (open circles) also. The dotted line represents the calculated regression for the tendency with $\Delta \dot{R}$ as a variable. Pnael *(c)*: $\Delta \dot{R}$ as function of the free parameter $k$. The dashed line is the theoretical approximation given by Equation (17), and the error bars are the difference between the average error of $\Delta \dot{r}$ and its maximum value, acording the upper right panel. Panel *(d)*: $\Delta \dot{r}$ as function of the difference between the calculated arrival-speed of the CME leading edge and the ambient solar wind. The dashed-line is the tendency-regression calculated for the data dispersion.



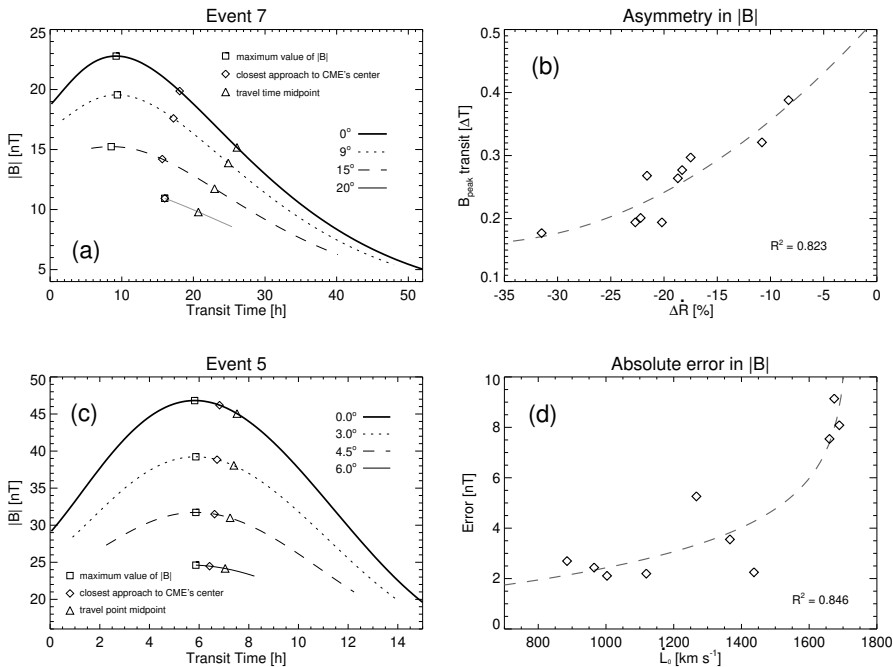

**Figure 7.** Effects of trajectory and expansion rate on synthetic profiles of magnetic field intensity, and absolute errors dependence on initial speed of CMEs. Panels *(a)* and *(c)* Synthetic profiles for different CME initial orientations for Events 7 (a) and 5 (c) during $\Delta T$. The different profiles correspond to initial CME trajectories deviated from the Sun-Earth line of sight. The open squares and open diamonds point out the maximum value of $|B|$ and the closest approach to CME center, respectively. Panel *(b)*: Transit of magnetic intensity peak, in terms of transit times, *vs* $\Delta \dot{R}$ for all events. Panel *(d)*: Absolute error for synthetic profiles of magnetic intensity. The calculated absolute errors as function of $\dot{L}_0$. In right panels the dashed lines are the performed regression for the data.

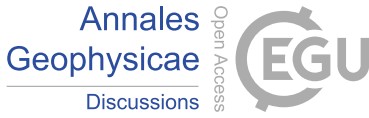

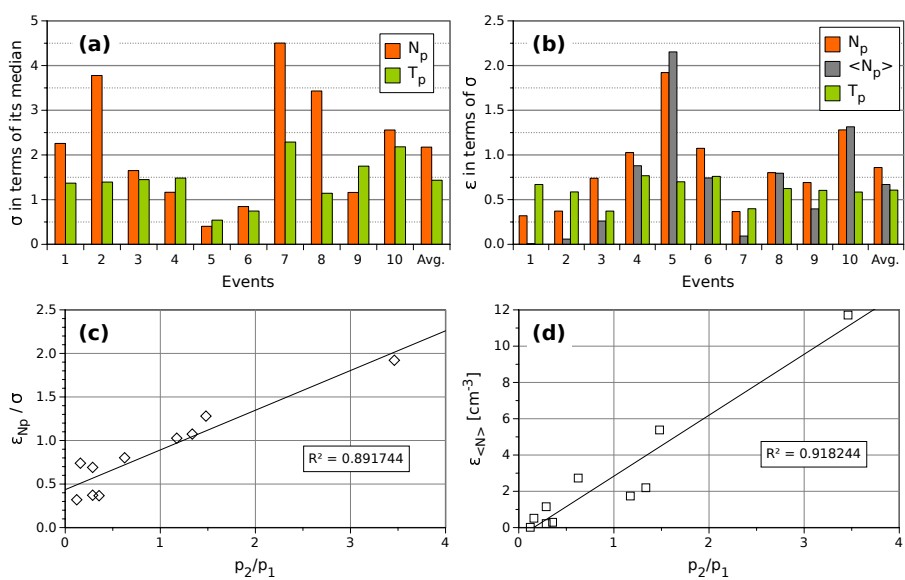

**Figure 8.** Histograms of the standard deviations and errors associated to density (orange) and temperature (green) of protons, and dispersion of density errors as functions of solar wind ram pressures quotient. Panel *(a)*: Standard deviation ($\sigma$) in terms of the associated median value ($< N_p >$), both calculated for *in situ* data during $\Delta T$. Panel *(b)*: Average errors ($\epsilon$) associated with our synthetic profiles in terms of their corresponding standard deviations. The rightmost bars in upper panels show the average values of $\epsilon$ and $\sigma$, respectively. Panel *(c)*: Density errors (open diamonds) in terms of their corresponding standard deviations *vs* ahead and behind ram pressures quotient. Panel *(d)*: Mean density errors (open squares) *vs* ahead and behind ram pressures quotient. Solid black lines in bottom panels are the corresponding regression tendencies, we also overplot the associated squared correlation coefficient.



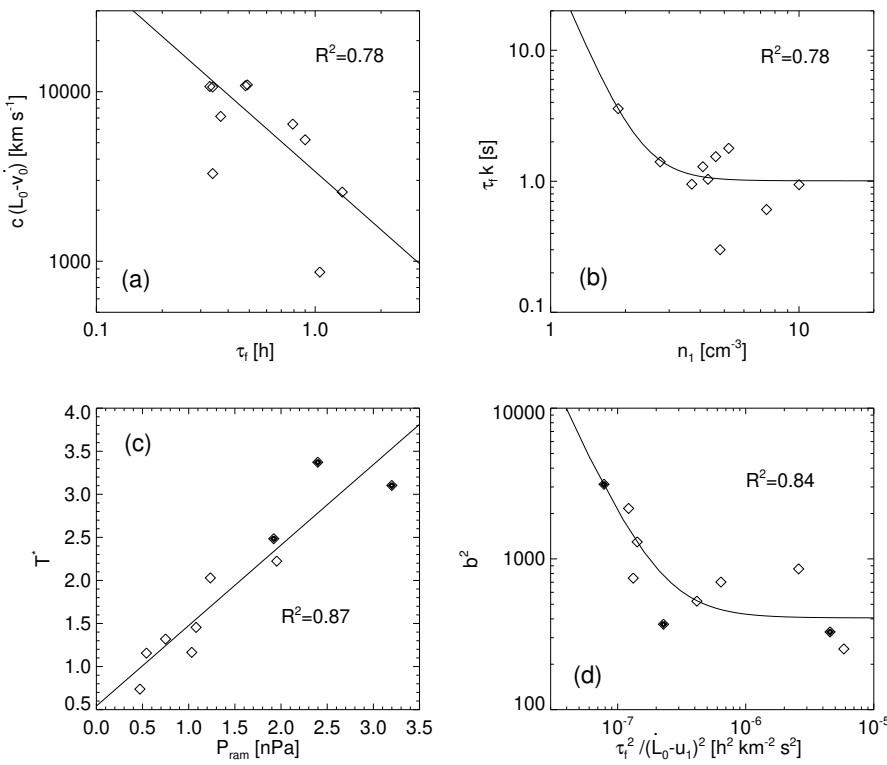

**Figure 9.** Empirical relations of free parameters and inputs. Open diamonds point out the data dispersion, the solid lines are the calculated regression tendencies, and we also present the squared correlation index between the data dispersion and the associated regression.



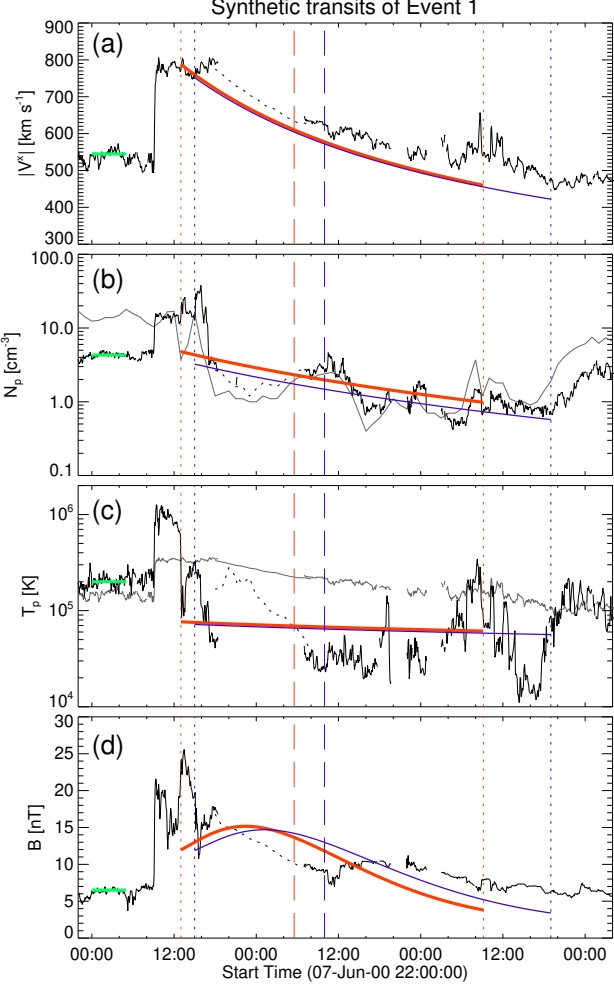

**Figure 10.** Synthetic transit profiles of event 1 calculated for different values of free parameters. The panels *(a)*, *(b)*, *(c)*, and *(d)* present the radial component of the solar wind speed ($|V^x|$), the density ($N_p$) and temperature ($T_p$) of protons and the magnetic field intensity ($B$), respectively. The red profiles were calculated with values of free parameters fixed by Equations (18) to (21); and the blue profiles are the same showed in Figure 4. Dotted and dashed vertical lines mark the calculated arrivals of the CME boundaries and center, respectively. Solid black lines are *in situ* measurements as extracted from NASA/GSFC's OMNI data set through OMNIWeb service. Green solid lines mark in situ solar wind used for calculations (see Table 1). Solid grey lines in $N_p$ and $T_p$ panels are the plasma beta (10 folded) and the expected proton temperature ($T_{exp}$) (Lopez, 1987), respectively. The free parameter values used to calculate the red synthetic transit were: $c = 18.29$, $k = 2.56$, $T^* = 2.16$, and $b = 23.0$.