# Peer review of "Development of a formalism for computing in situ transits of Earth-directed CMEs. Towards a forecasting tool II"

_Annales Geophysicae, 2019_

## Referee Comment (RC1) · Anonymous Referee #1 · 11 Feb 2020

This paper provides an empirical set of relationships that can be used to predict the in situ profiles of interplanetary coronal mass ejections (CMEs) as they transit through 1 AU (or at the Earth), namely predicting their speed, density, temperature and magnetic field strength time profiles. The method requires input information from coronagraph images and information on the upstream ambient solar wind conditions that the CME is propagating into. The method is tested on a number of example CMEs from the LASCO catalog against corresponding in situ data from the NASA-OMNI data set.

A difficulty I had reading the paper is that the formalism introduced in Section 2 is quite hard to follow, particularly in keeping connected to the physical picture being described,

unless previous papers by the same first author have been read. For example, I found myself quite confused on the meaning of the constants a and c, described simply as related to the inertia of the CME, until I had found and read Appendix A of Corona-Romero et al. (2017) (see reference list in the paper). While some referencing back for the finer details is of course appropriate, I recommend that the authors review and adjust Section 2 from the point of view of a reader encountering this formalism for the first time.

However, the methodology having been established, I found that the remainder of the paper describing the test cases to be a sensible approach and much more straightforward to follow. The method requires fine tuning of four free parameters to optimise the match to the in situ observations and the final section (4.2) of the paper explores how these might be estimated in a forecasting scenario by carrying out a parametric analysis of their relationship to the input observables within the test data set. I found these relationships somewhat tentative as possibly do the authors through their use of phrases such as "is somehow related" and "seems to be related". Nonetheless it would be very interesting to see how these relationships would perform in a test where the in situ data was not already known. Further limitations, if I have understood correctly, are that the methodology only applies to fast CMEs associated with solar flares that produce halo CMEs as viewed from the Earth. Overall I consider that the methods described have the potential to be a useful tool in space weather prediction for appropriate events and I am happy to support publication of this paper subject to minor revisions.

Suggestions and corrections:

Note that many of these refer to English language adjustments – I included these where I noticed them but this was not an exhaustive proof read which should still be carried out by the authors.

Line 21 – would it be worth a mention here that high speed streams are another source

and stating the relative importance of CMEs?

Line 93 – While -> while, and terminate previous line with a comma

Line 100 – roughly -> rough

Line 102 – believe -> believed

Line 107 – make clear that r is the distance from the Sun of the CME centre

Line 126 – on first reading I did not realise straightaway that TT_r was a single variable name.

Line 140 – another -> other

Line 146 – I don't think that the significance of the section of data described by the dotted black lines is ever explained.

Line 153 – TT as marked on the figure is actually the arrival time rather than the travel time – this needs to be rephrased.

Line 157 – list -> lists

Figure 2 – the yellow and green markers may need more emphasis

Line 168 – at in situ -> in situ

Line 181 – there is a missing reference indicated as ?

Line 190 – e.j. -> e.g., also at lines 195 and 200

Line 201 – relation -> a relation

Line 204 – I think you should immediately define b in the text which directly follows Equation 15; at the moment the definition comes a whole paragraph later.

Line 207 – there is a missing word before "by Gulisano", perhaps "reported".

Line 226 – the "accumulative magnetic flux" is discussed in a number of places in the

paper and would benefit from being clearly defined the first time, rather than just relying on the reference.

Table 1 – there is a missing reference in footnote b indicated by ?

Line 257 – currents -> streams

Line 266 – You use the term "aging" quite frequently – please explicitly define what you mean the first time rather than just relying on the reference.

Line 290 – tendency -> dependency, or relationship?

Line 335 – The Figure 7 -> Figure 7

Line 343 – The angular separation described here is not well defined – is it measured back at the Sun? I found the comparison to the work of Jian et al (2006) at the end of the paragraph interesting.

Line 376 – The first sentence of this paragraph is poorly worded – I was not clear on what point it is trying to make.

Line 386 – There is a broken reference to Figure 7.

Line 392 – even with the extra explanation here I was still not clear on the definition of the accumulated magnetic flux.

Line 412 – There is a reference to Figure 8 here – should it still be Figure 7?

Line 417 – The sentence starting "Of course" is poorly worded – I was not clear what it is trying to say.

Line 430 – derived into -> resulted in

Line 438 – the Figure 8 -> Figure 8

Line 447 – CMEs -> the CMEs

Line 469 – There is a missing reference indicated by ?

[Figure]

Line 534 – For -> By

Line 546 – Linquist -> Lundquist (also at line 626); not longer -> no longer

Line 548 – among of others -> among others

Line 565 – it is an -> it is a

Line 577 – simpler -> simple

Line 581 – help in -> help to

Line 596 – We realized -> We note (also at line 633)

Line 617 – Regarding of -> Regarding the

Line 622 – rephrase to "an assumption that contrasts"

Line 643 – for construction -> by construction

Line 657 – of CMEs -> which require to be specified for each CME (?)

Line 660 – of CMEs -> of the CMEs

Line 667 – that relates -> relates

Line 722 – empiric -> empirical

Line 725 – our we -> we

Line 732 – occurred -> which occurred

Line 740 – a curved-like -> curved-like

Line 756 – redaction -> production

Figure 6(c) – In legend, theoric -> theoretical

---

## Referee Comment (RC2) · Anonymous Referee #2 · 12 Feb 2020

This paper discusses a scheme to forecast the 1 AU properties of coronal mass ejections, at least those that are relatively fast (so the "piston approximation" can be assumed) land aunched near central meridian, that is based on an analytical model and empirical relationships. The major part of the paper describes an exhaustive comparison of the predicted and observed CME parameters. I did perhaps find this discussion overlong, given that this is just one model with limitations, such as the CMEs should be launched near central meridian, and am not sure whether it will be of interest beyond a limited audience. Also, I wasn't really sure what I learned from reading the paper. However, the paper is generally clearly written other than some minor English errors and overall, I recommend publication subject to consideration of a few minor points.

[Figure]

Line 16: Is Moldwin really the correct reference for the NSWPSP? Is there a specific reference for the Plan?

Line 22: CIRs and high speed streams can also be responsible for geomagnetic effects of course.

Line 23: Also give a couple of references to refereed journal articles since the reader might not have access to Howard's book.

Line 42: I don't believe this 1990 paper is the correct reference for the Wang-Sheeley-Arge model; Arge only arrived on the scene later (early 2000s?).

This is the third study, but numbered "II". Maybe an additional phrase could be added to the title to describe what aspect this paper discusses, e.g. "... II: Test of the model against observations" or whatever's appropriate.

Line 89: "Rising phase duration" of what aspect of the flare, e.g. soft X-rays, or something else? This is finally clarified in line 730.

Line 102: believed

In Figure 1 (left), theta is indicated as the full angular width, not the semi-angular width. Perhaps the dashed arc should extend only to the horizontal line?

Line 145: This isn't the simplest of CMEs, with structures in the CME. Is there a particular reason it was chosen as an example? This also applies to many of the other events. Later it is mentioned that the modeling doesn't predict these details.

Line 155: Give the CME parameters and mention which CME is associated with this in situ CME.

In Figure 2, why are certain sections of the data (inside the CME boundaries) shown as dashed rather than solid lines? This isn't mentioned in the caption.

Figure 2 caption, line 4: The CME boundary lines aren't dotted, but are short dashes

vs. long dashes for the center line.

Caption, line 6: Lopez (1987) doesn't actually discuss the expected temperature. I believe the expected temperature was introduced by Richardson and Cane 1993 (DOI: 10.1029/93JA01466) and 1995 (DOI: 10.1029/95JA02684), who used a Tp vs. V fit from Lopez and the observed solar wind speed to calculate Texp, which was then compared with the observed Tp. Lopez is also referenced several other times in the paper in this way.

Line 174: Multiple coronagraph observations are not available for this event (the STEREOs were launched in 2006), so the speed can't be "fixed" in this way.

Line 181: Missing reference at "?"? 12% seems too high for the alpha component of most CMEs, e.g., Figure 5 of doi:10.1029/2004JA010598, and enhanced He/p isn't a consistent signature of CMEs. In any case, there should be observations for the events of interest to check this ratio.

Line 195: "and many others" isn't useful for the reader.

Line 226: Explain briefly how the accumulative magnetic flux is calculated. The reader might expect this to be a sum of the measured intensity or something similar and hence continue to increase with time, but clearly this is not the case.

Table 1: How are the CME associations with the in situ CMEs made? E.g., are they well-established in the literature or inferred by the authors? Obviously, they have to be correct, otherwise the analysis breaks down, so this deserves a few words of explanation.

Footnote b: Is there a missing reference or a query at the "?"?

Line 256: Not quite sure what is meant by "when we neglect the events affected by interacting currents of solar wind".

Figure 4 caption: "Dotted red lines . . ." mentioned twice.

Line 376: I don't understand "The certainest on the trajectory of a CME as a whole may help to know the transit of the CME boundaries as well as the closest approach to the CME center."

Line 378: do not necessarily

Line 386: Undefined figure?

Line 430: Not sure what "conditions that derived into the short bars shown in the histogram" means.

Line 469: Missing reference?

Line 673: The curve is only determined by one point, so how reliable/useful is this fit?

Line 717: Of course this requires other information, for example, on Bz in the sheath which isn't within the capabilities of such models.

Line 755: "redaction" doesn't seem the correct word -production?

---

## Author Comment (AC1) · 18 Mar 2020

The authors express thanks to the Referee for the constructive suggestions to our manuscript. We addressed all the "English language adjustments" as you requested and updated the Figures when necessary. Below, we describe our approach to address the technical issues that were raised.

Line 21: We include high-speed streams as source for space weather phenomena.

Line 153: We modified the text to clarify this point. Now it reads: "We mark the CME arrival time by a vertical dotted red line on the left side of the panels in Figure 2, and the

corresponding TT would be the time lapse in between the arrival and detection times."

Figure 2: We modified the figure's caption to improve the description of the yellow and green lines.

In the original version we wrote:

"Green solid lines mark in situ solar wind used for calculations and solid yellow lines mark solar wind behind the CME (see Table 1)."

This now reads:

"The green solid lines, on the left side of all panels, mark the in situ solar wind values used as inputs for our calculations; and solid yellow lines, in $|V\hat{x}|$ and $N\_p$ panels, mark speed and proton density values of the solar wind behind the CME (see Table 1)."

Line 226: We added a few sentences to briefly describe the way we calculate the accumulative magnetic flux. The new reads:

"In order to compute it, we integrate the (poloidal) magnetic field component, that is simultaneously perpendicular to the propagation direction and axial component, along the spacecraft transit inside the CME. For this purpose, we use the maximum variance technique to infer the reference frame of the CME magnetic field and use the magnetic coordinate of largest variance to calculate the accumulated magnetic flux as a function of time. It is important to note that this extremum gives an estimation for the time of closest approach to the magnetic center inside the CME."

Line 266: We added a short description for the "aging effect".

In the previous version the text read:

"It is important to note that all of our synthetic speed-profiles reproduce the speed-decreasing tendency called aging (Osherovich et al., 1993), commonly associated with the CME expansion. The aging effect is also present in the in situ data; however, it varies from one event to another; a condition that is easily observed in synthetic

profiles..."

In our revised version the text reads:

"It is important to note that all of our synthetic speed-profiles reproduce the monotonic speed-decreasing tendency called aging (Osherovich et al., 1993), commonly associated with the CME expansion. The aging effect refers to the change in the CME characteristics seen in in situ measurements during the spacecraft transit across the CME structure; such a change is mainly due to CME expansion. The aging effect is also present in the in situ data; however, it varies from one event to another; a condition that is easily observed in synthetic profiles..."

Line 392: We expect that corrections relative to "Line 226" would help to clarify this point.

Finally, in your general description about the manuscript, you make two comments that are particularly interesting:

I) Section 2 is quite hard to go, and that II) the meaning of constants "a" and "c" may be confusing.

To address the first point, we propose to add the following sentences as a brief introduction to the model, at the beginning of Section 2.

Previously, the text read:

"The CME trajectories calculated by the piston-shock model have two phases: a short interval of constant speed followed by a period where the CME speed asymptotically approaches the speed of the solar wind. Previous studies suggested that the first phase ends around 30 R , hence the deceleration phase dominates CME propagation up to the orbit of the Earth (d â£Ṯ = 1 AU ) (Corona-Romero et al., 2013, 2015)."

The proposed change is:

"The piston-shock model is an analytical approach that assumes the CME as a piston,

driving a shock wave during a finite lapse of time. The model simultaneously solves the CME leading edge (L_dot) and shock front positions. In order to calculate L_dot the model assumes conservation of both linear momentum and mass, in the interaction between the CME and surrounding solar wind. The CME trajectories calculated by the piston-shock model have two phases: a short interval of constant speed followed by a period where the CME speed asymptotically approaches the speed of the solar wind. The first phase occurs during its injection into the interplanetary medium, as long as the CME is provided with external energy. Once the external energy supply is exhausted, the interaction with the ambient solar wind decelerates the CME, which tends to equalize its surrounding solar wind speed. Previous studies have suggested that the first phase ends around 30 Ro , hence the deceleration phase dominates CME propagation up to the orbit of the Earth (dâŁʈ = 1 AU ) (Corona-Romero et al., 2013, 2015)."

To address the second issue, we propose to add a new sentence when defining constant "a":

In the original version the text read:

"...while c is treated as a free parameter to match the calculated arrival time with its in situ registered counterpart."

In this new version the text reads:

"...while c is treated as a free parameter to match the calculated arrival time with its in situ counterpart. In the piston-shock model the constants "a" and "c" define the CME injection values of speed and density relative to the solar wind's values, respectively."

Again, we would like to thank the referee for their valuable and constructive comments. We appreciate them taking the time to review the manuscript so carefully.

---

## Author Comment (AC2) · 18 Mar 2020

The authors appreciate your positive and constructive comments.

We addressed all the grammatical errors you identified. We also added the missing citations as well correcting the issues with the Figures and Captions. In the following paragraphs, we address each of your remaining remarks.

Line 89: Your comment is correct, we modified the text to include a brief description for tau_f.

In last version the text read:

"Additionally, u_1 is the in situ solar wind bulk speed and tau_f the rising phase duration (Zhang and Dere, 2006) of the associated solar flare."

Now the text reads:

"Additionally, u_1 is the in situ solar wind bulk speed and tau_f the rising phase, i.e. the period between the maximum and start times of the associated solar flare's X-ray flux (see Zhang and Dere, 2006, for further details on rising phase of solar flares)."

Line 145: Yes, there is a particular reason to use the events listed in our Table 1; these are the events that fulfill most of our piston-shock model assumptions. These "well behaved" events were selected from Table 1 of the previous work by Corona-Romero et.al. [2017]. Please see this reference for further detail on the events. In our draft, our criteria of selection included the error associated with computed travel times and arrival speeds of CMEs. Of all the cases, event 1 was the one with the lowest errors, and the reason for which we used it to demonstrate our methodology.

Line 155: We modified the text to clarify the parameters we are using as inputs.

In the previous version the text read:

"The values of L 0 and LÌĞ 0 we used were the reported initial position and speed by CME LASCO Catalog (Yashiro et al., 2004; Gopalswamy et al., 2009), respectively."

The text now reads:

"We used the initial position, at the first appearance in C2, and the linear speed reported by CME LASCO Catalog (Yashiro et al., 2004; Gopalswamy et al., 2009) as inputs for the values of L_0 and LÌĞ_0, respectively."

We do not state that initial position (L_0) and speed (LÌĞ_0) correspond to the CME leading edge position and speeds at in situ measurements for several reasons. First, at this point in the text, such a clarification may confuse the reader. Additionally, we believe it could be somehow repetitive, since in Section 2 we make it clear that L and

L_dot describe the position and speed of CME's leading edge. However, if the Referee feels that the text requires further clarification on this issue, we would gladly add few sentences to address it.

Line 181: According to Zurbuchen & Richardson [2006] (see Table 1), the enhancement (>8%) of alpha particles vs protons is the primary plasma-composition in-situ (at 1 AU) signature for CME detection. The authors agree that assuming an average value of 12% of alpha particles for all events might induce unnecessary uncertainties; nevertheless we believe that such a value does not invalidate our results. We have provided [attached to this letter] the in situ values of alpha particle rate during the in situ transit of the events studied, should the referee like to check the values for each one.

Line 673: Perhaps the authors did not adequately express their ideas in the Section 4.2. The equation (18) is not determined by one point, but by 10; which is the number of our analyzed events and, thus, the number of available data points. This also applies for the rest of the empirical relations.

Line 717: We agree, and have modified the text to address this. In previous version of the manuscript it read:

"The capability to simultaneously forecast in situ transits of CMEs, the geoeffectiveness, associated forward shocks, and plasma sheaths is of great interest for space weather purposes, since more intense geomagnetic storms are triggered by such phenomena (Ontiveros and Gonzalez-Esparza, 2010, and references therein). If this formalism is shown to be robust under a range of conditions, it can lead to an important operational tool for space weather, particularly for those scenarios when the response time is of importance, like early warning systems..."

The text now reads:

"The capability to simultaneously forecast in situ transits of CMEs, the geoeffectiveness, associated forward shocks, and plasma sheaths is of great interest for space

weather purposes, since more intense geomagnetic storms are triggered by such phenomena (Ontiveros and Gonzalez-Esparza, 2010, and references therein). However, such a goal also requires additional information, such as the magnetic field within the sheath regions, which is not within the capabilities of such models yet. Nevertheless, if this formalism is shown to be robust under a range of conditions, it can lead to an important operational tool for space weather, particularly for those scenarios when the response time is of importance, such as early warning systems..."

Again, we would like to thank the referee for their valuable and constructive comments. We appreciate them taking the time to review the manuscript so carefully.
* * *
OMNI2

**Fig. 1.** Event 1

**Fig. 2.** Event 2

OMNI2

**Fig. 3.** Event 3

[Figure]

[Figure]

**Fig. 4.** Event 4

[Figure]

**Fig. 5.** Event 5

**Fig. 6.** Event 6

[Figure]

OMNI2

**Fig. 7.** Event 7

[Figure]

Fig. 8. Event 8

OMNI2

**Fig. 9.** Event 9

---

## Author Response (AR1)

Referee #1

The authors express thanks to the Referee for the constructive revision of our manuscript. We attended all the "English language adjustments" in the way you requested. Thus, in next paragraphs, we describe our approach to address those no-language points that you listed.

Line 21: We include high speed streams as source for space weather phenomena.

Line 153: We modified the text to clarify this point. Now it reads: "We mark the CME arrival time by a vertical dotted red line on the left side of the panels in Figure 2, and the corresponding TT would be the time lapse in between the arrival and detection times."

Figure 2: We modified the figure's caption to improve the yellow and green lines description.

In the original version it was: "Green solid lines mark in situ solar wind used for calculations and solid yellow lines mark solar wind behind the CME (see Table 1)."

Now it is: "The green solid lines, on the left side of all panels, mark the in situ solar wind values used as inputs for our calculations; and solid yellow lines, in $|V^x|$ and $N\_p$ panels, mark speed and proton density values of the solar wind behind the CME (see Table 1)."

Line 226: We added a new sentences to briefly describe the way we calculate the accumulative magnetic flux. The new lines are: "In order to compute it, we integrate the (poloidal) magnetic field component, that is simultaneously perpendicular to the propagation direction and axial component, along the spacecraft transit inside the CME. For this purpose, we use the maximum variance technique to infer the reference frame of the CME magnetic field, and uses the magnetic coordinate of largest variance to calculate the accumulated magnetic flux as function of time. It is important to note that this extremum gives an estimation for the time of closest approach to the magnetic center inside the CME."

Line 266: We added a short description for the "aging effect".

In the previous version it read: "It is important to note that all of our synthetic speed-profiles reproduce the speed-decreasing tendency called aging (Osherovich et al., 1993), commonly associated with the CME expansion. The aging effect is also present in the in situ data; however, it varies from one event to another; a condition that is easily observed in synthetic profiles..."

In our revised version it reads: "It is important to note that all of our synthetic speed-profiles reproduce the monotonic speed-decreasing tendency called aging (Osherovich et al., 1993), commonly associated with the CME expansion. The aging effect refers to the change in the CME characteristics seen in in situ registers during the spacecraft transit accross the CME structure; such a change is mainly due to the CME expansion. The aging effect is also present in the in situ data; however, it varies from one event to another; a condition that is easily observed in synthetic profiles..."

Line 392: We expect that corrections relative to "Line 226" would help to clarify this point.

Finally, in your general description about the revision, your commented two issues that are of special interest for us:
I) Section 2 is quite hard to go, and that
II) the meaning of constants "a" and "c" may be confusing.

To address the first issue, we propose to add the next sentences as an brief intro to the model, at the beginning of Section 2.

In the previous version it read: "The CME trajectories calculated by the piston-shock model have two phases: a short interval of constant speed followed by a period where the CME speed asymptotically approaches the speed of the solar wind. Previous studies suggested that the first phase ends around 30 R , hence the deceleration phase dominates CME propagation up to the orbit of the Earth ($d_\oplus = 1\,AU$) (Corona-Romero et al., 2013, 2015)."

Now it reads: "The piston-shock model is an analytical approach that assumes the CME as a piston that drives a shock wave during a finite lapse of time. The model solves simultaneously the CME leading edge (L_dot) and shock front positions. In order to calculate L_dot the model assums conservation of both, linear momentum and mass, in the interaction between the CME and surrounding solar wind. The CME trajectories calculated by the piston-shock model have two phases: a short interval of constant speed followed by a period where the CME speed asymptotically approaches the speed of the solar wind. The first phase occurs, during its injection into the interplanetary medium, as long as the CME is provided with external energy. Once the external energy supply is exhausted, the interaction with the ambient solar wind decelerates the CME, which tends to equalize its surrounding solar wind speed. Previous studies suggested that the first phase ends around 30 Ro , hence the deceleration phase dominates CME propagation up to the orbit of the Earth ($d_\oplus = 1\,AU$) (Corona-Romero et al., 2013, 2015)."

To address the second issue we propose to add a new sentence when defining constant "a":

In the original version it read: "...while c is treated as a free parameter to match the calculated arrival time with its in situ registered counterpart."

In this new version it reads: "...while c is treated as a free parameter to match the calculated arrival time with its in situ registered counterpart. In the piston-shock model the constants "a" and "c" define the CME injection values of speed and density relative to the solar wind's ones, respectively."

Referee #2

The authors appreciate your positive revision and constructive comments.

We addressed all the English errors you remark. We also solved the missing citations as well the Figures and Captions issues. In the next paragraphs, we describe the way we addressed the rest of your remarks.

Line 89: Your comment is correct, we modified the text to include a brief description for tau_f.

In last version it was: "Additionally, u_1 is the in situ solar wind bulk speed and tau_f the rising phase duration (Zhang and Dere, 2006) of the associated solar flare."

Now it is: Additionally, u_1 is the it in situ solar wind bulk speed and tau_f the rising phase, i.e. the period between the maximum and start times of the associated solar flare's X-ray flux (see Zhang and Dere, 2006, for further details on rising phase of solar flares).

Line 145: Yes, there is a particular reason to use the events listed in our Table 1; the reason is that those events fulfill most of our piston-shock model assumptions. These "well behaved" events were selected from Table 1 of the previous work by Corona-Romero et.al. [2017], please see the reference for further detail on the events. In our draft, our criteria of selection included the error associated with computed travel times and arrival speeds of CMEs. Between all cases, the event 1 was the one with the lowest errors, reason for which we used it to show our methodology.

Line 155: We modified the text to clarified the parameters we are using as inputs.

In the previous version it read: "The values of $L_0$ and $\dot{L}_0$ we used were the reported initial position and speed by CME LASCO Catalog (Yashiro et al., 2004; Gopalswamy et al., 2009), respectively."

Now it reads: "We used the initial position, at the first C2 appearance, and the linear speed reported by CME LASCO Catalog (Yashiro et al., 2004; Gopalswamy et al., 2009) as inputs for the values of $L_0$ and $\dot{L}_0$, respectively."

The authors would like do not comment that initial position ($L_0$) and speed ($\dot{L}_0$) correspond to the CME leading edge position and speeds at in situ measurements. First, in this position of the text, such a clarification may compromise the sequence of the ideas. Additionally, we believe it could be somehow repetitive, since in Section 2 we think we made it clear enough that L and L_dot describes the position and speeds of CME's leading edge. However, if the Referee insists in the necessity of further clarification on this issue, we would gladly add few sentences to address it.

Line 181: According to Zurbuchen & Richardson [2006] (see Table 1), the enhancement (>8%) of alpha particles vs protons is the first plasma-composition in-situ (at 1 AU) signature for CME detection. The authors agree that assume an average value of 12% of alpha particles for all events might induce unnecessary uncertainties; nevertheless we believe that such a value does not invalidate our results. We attached the in situ values of alpha particle rate during the in situ transit of the studied events in the case the Referee would like to check the values for each case.

Line 673: Perhaps the authors did not adequately express their ideas in the Section 4.2. The equation (18) is not determined by one point, but by 10; which is the number of our analized events and, thus, the number of available data points. This also applies for the rest of the empirical relations.

Line 717: We agree. We modified the text.

[revised manuscript text omitted]